# Intervention Strategies to Control *Campylobacter* at Different Stages of the Food Chain

**DOI:** 10.3390/microorganisms11010113

**Published:** 2023-01-01

**Authors:** Khaled Taha-Abdelaziz, Mankerat Singh, Shayan Sharif, Shreeya Sharma, Raveendra R. Kulkarni, Mohammadali Alizadeh, Alexander Yitbarek, Yosra A. Helmy

**Affiliations:** 1Department of Animal and Veterinary Science, College of Agriculture, Forestry and Life Sciences, Clemson University, Clemson, SC 29634, USA; 2Department of Pathobiology, Ontario Veterinary College, University of Guelph, Guelph, ON N1G 2W1, Canada; 3Department of Population Health and Pathobiology, College of Veterinary Medicine, North Carolina State University, Raleigh, NC 27606, USA; 4Department of Animal Science, McGill University, Montreal, QC H9X 3V9, Canada; 5Department of Veterinary Science, College of Agriculture, Food, and Environment, University of Kentucky, Lexington, KY 40546, USA

**Keywords:** chicken, *Campylobacter*, vaccine, feed additives, probiotics, prebiotics, synbiotics, bacteriophages, bacteriocins, organic acids, essential oils, small molecules, short-chain fatty acids

## Abstract

*Campylobacter* is one of the most common bacterial pathogens of food safety concern. *Campylobacter jejuni* infects chickens by 2–3 weeks of age and colonized chickens carry a high *C. jejuni* load in their gut without developing clinical disease. Contamination of meat products by gut contents is difficult to prevent because of the high numbers of *C. jejuni* in the gut, and the large percentage of birds infected. Therefore, effective intervention strategies to limit human infections of *C. jejuni* should prioritize the control of pathogen transmission along the food supply chain. To this end, there have been ongoing efforts to develop innovative ways to control foodborne pathogens in poultry to meet the growing customers’ demand for poultry meat that is free of foodborne pathogens. In this review, we discuss various approaches that are being undertaken to reduce *Campylobacter* load in live chickens (pre-harvest) and in carcasses (post-harvest). We also provide some insights into optimization of these approaches, which could potentially help improve the pre- and post-harvest practices for better control of *Campylobacter*.

## 1. Introduction

*Campylobacter* spp. are the leading bacterial cause of food-borne illness globally [1]. *Campylobacter jejuni* and *C. coli* are closely related species [2], both involved in food-borne illness, with *C. jejuni* being the most commonly isolated species [3]. The incidence of campylobacteriosis is increasing in many countries, and it is hyperendemic in children under 5 years of age in developing tropical regions [4,5]. *C. jejuni* infections are often associated with gastroenteritis and approximately 30% of acute enteritis cases develop a severely debilitating irritable bowel syndrome with a disease fatality rate that is estimated to be about 5 deaths per 100,000 cases [4,6,7]. Furthermore, some strains of *Campylobacter* have been associated with several autoimmune disorders, most notably Guillain-Barré syndrome [8,9]. It is estimated that approximately 40% of the reported Guillain-Barré syndrome cases are attributed to this bacterium [10]. While severe complications are not as prevalent in developed countries, *C. jejuni* remains one of the most reportable bacterial pathogens of food safety concern in Europe and North America [11,12]. According to the latest USDA-economic research service report, the annual healthcare costs incurred by foodborne diseases in the US are estimated to be about $15 billion, of which $1.6 billion caused by *Campylobacter* species [13]. Data from the Centers for Disease Control and Prevention (CDC) indicates “there are about 1.3 million cases of *Campylobacter* infection each year in the US alone” [14].

Although domestic mammals and environmental contamination may be sources of infection, chickens are the main source of infection in humans, with 50–80% of the reported cases of campylobacteriosis in Europe being attributed to consumption of poultry products contaminated with this microbe [15,16,17]. Chickens are considered a natural reservoir for *C. jejuni*, as their intestinal tract offers an optimal biological niche for the survival and proliferation of this bacterium [18]. Chicks often become colonized by 2–3 weeks of age; however, they are largely asymptomatic post-colonization [19,20,21]. Contamination of chicken carcasses with the gut content of *Campylobacter*-infected chickens during processing at slaughter plants poses a threat to human health [22]. Therefore, three general disease control strategies have been put in place to reduce *Campylobacter* burden in poultry encompassing pre- and post-harvest intervention checkpoints. The pre-harvest checkpoints are (a) reducing environmental exposure to *Campylobacter* via biosecurity measures, (b) increasing avian host defense through vaccination, and (c) using antibiotic alternative products to reduce or clear infection load in chickens. The post-harvest checkpoints include (a) slaughter plants cleaning and sanitation, (b) carcass decontamination, and (c) eggshell sanitation. In this context, several approaches have been investigated for control of *C. jejuni* colonization and transmission in poultry, including on-farm biosecurity measures and the use of immune-based strategies, such as vaccines and feed additives (prebiotics, probiotics, essential oils, bacteriophages, etc.) [23,24,25,26]. However, none of these strategies, by themselves, can eliminate the microbial carriage and the risk of shedding and transmission. Therefore, other complementary control measures are needed to reduce *Campylobacter* load on chicken carcasses. This article aims to shed light on various control measures being investigated to reduce the risk of *Campylobacter* transmission from poultry products to humans, including pre- and post-harvest control programs (Figure 1).

## 2. Pre-Harvest Control Measures (On-Farm Control)

Biosecurity measures and intervention strategies that have been applied to reduce *Campylobacter* burden in poultry flocks.

### 2.1. On-Farm Biosecurity Measures

Ensuring newly hatched chicks are protected from infection with *Campylobacter* requires identification of potential sources for transmission [27]. Horizontal transmission from older flocks is considered the most common route of *Campylobacter* infection in juvenile chickens [28], especially in farms lacking sophisticated isolation between poultry houses (Figure 2). A minimum infectious dose as little as 50 organisms can colonize the chicken gut [29]. Once *Campylobacter* reaches the intestinal tract of chickens, it colonizes the small intestine and cecum in high numbers [20]. Infected chickens carry and shed this bacterium in their feces and subsequently spread the infection to the entire flock without developing clinical disease [20] The presence of many poultry houses and older flocks on site is, indeed, amongst the greatest risk factors for flock colonization by *Campylobacter* [30].

Contamination of the surrounding environment with fecal droppings and contaminated water from the adjacent farm, rodents, insects, and wild animals also considered potential sources of *C. jejuni* infection in broilers [31,32,33,34,35]. Nonetheless, even with establishing single species farms with secured access points, traffic control and a minimum of 200 M distance between the poultry houses, these measures are still insufficient to prevent the transmission of *C. jejuni* into the poultry houses [19].

In addition to horizontal transmission from neighboring flocks, one of the other significant sources of infection in production facilities can be from the lack of proper litter treatment between flock turnover or recycling old litters from a previous flock which is quite common in some countries [36].

While the horizontal route plays a major role in *Campylobacter* transmission, vertical transmission is unlikely [34,37]. A number of studies have reported infrequent detection of *C. jejuni* in chicken eggs [38,39]; however, whether it is due to internal eggshell contamination from the hen’s reproductive tract or external eggshell contamination with feces of infected chicken remains controversial. In an experimental trial, *C. jejuni* was detected in 20% of Specific Pathogen Free eggs (SPF) 3 h following the exposure to *C. jejuni*-contaminated wood shavings. In another study, inoculation of *C. jejuni* into the air space of embryonated SPF eggs resulted in high embryonic mortality, with 87% of embryos died on the second day post incubation [40]. Nonetheless, there is as yet no evidence whether heavy contamination of eggshell with *Campylobacter* from environmental sources would lead to similar outcomes.

*Campylobacter* transmission can also be attributed to human activity on poultry farms; contaminated clothes, skin, and boots of farmworkers and transport crates, either due to biosecurity breaches during the process of partial flock depopulation (thinning) or insufficient implementation of strict biosecurity measures, may contribute to the transmission of *Campylobacter* from the external environment into poultry houses [20,33,41]. Partial flock depopulation is a common practice in many European countries where a portion of a flock is removed and sent for slaughtering before the final slaughter age. This process was found to be associated with an increased risk of *Campylobacter* introduction into the broiler house [42]. In a recent survey in Ireland, an increase in prevalence of *Campylobacter* infection was observed due to the thinning process, where *Campylobacter* was detected in 38% of the neck skin samples of chickens removed early for slaughtering compared to 67% in the remainder flock [43].

It is widely believed that raising chickens in modern state-of-the-art production systems and full implementation of biosecurity measures would significantly tackle *Campylobacter* infections. Yet, *Campylobacter* remains a challenge despite the fact that such measures are being undertaken in European countries, such as the UK, and New Zealand [41,44]. As a result, novel on-farm mitigation strategies and complementary approaches should be applied.

### 2.2. Immune-Based Strategies to Control Campylobacter Colonization in Poultry Flocks

#### 2.2.1. Vaccines

Vaccine development for poultry has long been pursued with the hopes of reducing *Campylobacter* colonization and ultimately the incidence of human disease by reducing *Campylobacter* load on chicken carcasses [45,46,47]. *Campylobacter* colonization in young chickens is generally not seen until 2–3 weeks post-hatching, probably due to the availability of maternally derived antibodies to this bacterium [48]. Indeed, serum IgY antibodies against *C. jejuni* are routinely found in breeder hens, and maternal IgY antibodies are detected in serum of their progenies up to about 14 days post-hatch. However, following this period of passive immunity, chicks become receptive to colonization, especially in cases where *Campylobacter* is present in older flocks and the production facility environment [48]. The immune responses following infection do not appear to limit *Campylobacter* colonization by the age when broiler birds are slaughtered [20].

Due to the asymptomatic nature of *Campylobacter* colonization in poultry, with no associated reduction in growth or production efficiency, the poultry industry has not significantly pushed for the development of *Campylobacter* vaccines. However, the burden of disease in humans is substantial with billions of dollars lost annually due to medical care attributed to the disease [1]. Despite over two decades of attempts to develop a successful *Campylobacter* vaccine in broilers, successful commercial vaccines are not currently available. Many vaccination trials have been carried out to date, employing a variety of methodologies, such as whole-cell or subunit vaccinations, microorganism-vectored vaccines as well as nanoparticles vaccines (Table 1).

##### Whole Cell Vaccines (WCV)

Vaccines made up of killed whole-cell or attenuated bacteria have been trialed for the control of *Campylobacter* infections in chickens, however, they have demonstrated minimal effectiveness. The use of killed bacteria, which lack the ability to replicate and colonize chicken’s gut, has been tested against *Campylobacter* colonization in poultry. In three vaccine trials [46], repeated oral administration of formalin-inactivated *Campylobacter* cells to chickens, followed by exposure to *Campylobacter*, was reported to result in a significant elevation in serum and bile *C. jejuni*-specific IgA titers in the vaccinated birds and a variable reduction in *Campylobacter* colonization ranging from 16 to 93% compared to the non-vaccinated control chickens. Intraabdominal injection of attenuated *Campylobacter* cells paired with flagellin protein and 3 other *Campylobacter* antigens resulted in a 2- log_10_ reduction in cecal *Campylobacter* loads; however, when administered orally, no effect was observed. Similarly, Ziprin et al. [50] have attempted to use viable non-colonizing *Campylobacter* cells, manufactured by mutating a highly infectious strain of *Campylobacter*, however, the vaccine failed to provide protection against colonization. Additionally, Noor et al. [63] found that chicks inoculated *in ovo* and boosted orally with WCV after hatching had a significant immunological response, with *C. jejuni*-specific IgY, IgA, and IgM antibodies detectable in serum and IgA in intestinal contents and bile. On the other hand, Glünder et al. [64] found that despite the presence of specific antibodies in chicken serum following subcutaneous immunization with formalin-inactivated *C. jejuni* and complete Freund’s adjuvant, a little reduction in colonization was observed after a homologous challenge, while no effect was observed after a heterologous inoculation.

##### Subunit Vaccines

The immunodominant antigen of *Campylobacter*, flagellin, was used to create the first subunit vaccination in chickens [45]. Flagellin is regarded as a major virulence factor of *Campylobacter* as it plays a crucial role in bacterial motility, adhesion and colonization [65]. It was reported that the administration of a *C. jejuni* flagellin subunit vaccine to 18-day-old chicken embryos resulted in a significant production of serum IgY and IgM antibodies, however, it failed to stimulate intestinal IgA production, and was also ineffective at protecting chickens against *Campylobacter* colonization [66]. Some researchers have proposed the use of purified native flagellin for subunit vaccination. For example, Neal-McKinney et al. [53] found that birds vaccinated with flagellin coupled with the Montanide adjuvant had a high specific IgY antibodies and a 3 log_10_ decrease in intestinal *Campylobacter* count. Despite the encouraging results, flagellin cannot be employed for large-scale chicken immunization due to: (1) variations in flagellin across *Campylobacter* strains could potentially lead to failure of vaccines to adequately confer cross protection against the variety of strains that can colonize broilers, (2) anti-flagellin antibodies may target non-surface-exposed epitopes, and thus fail to mediate *Campylobacter* clearance, and (3) anti-flagellin antibodies may recognize glycosylated residues distributed over the surface of flagellin, allowing *Campylobacter* to elude the host’s immune system [67,68,69].

The potential of *Campylobacter* capsular polysaccharide-diphtheria toxoid conjugated subunit vaccine to protect against *Campylobacter* has also been evaluated in both humans and animals [54,70]. Despite the ability of this vaccine to afford protection against *Campylobacter* in mice and Aotus monkey models, it resulted in minimal reduction (0.64 log_10_) of cecal colony forming units (CFUs) of *C. jejuni* in broiler chickens. Nonetheless, these results provide insightful information that successful *Campylobacter* vaccines for chickens and humans may differ in their antigenic targets.

Generally, several factors are associated with the failure of conventional vaccines to provide effective protection against *Campylobacter* infection, foremost of which is the high genetic diversity across *Campylobacter* serotypes [71], and the fact that chickens are exposed to several different strains of *Campylobacter* over their lifespans. Overall, although several studies have demonstrated the development of specific antibody immune responses with whole-cell and subunit vaccines, these vaccines have largely been inconsistent and unsuccessful in chickens [64,66,72], and the focus on conventional vaccine development has largely shifted towards the development of recombinant and nanoparticles vaccines.

##### Recombinant Vaccines

These vaccines are manufactured through recombinant DNA technology, whereby antigen-specific DNA is inserted into bacterial or mammalian cells which subsequently replicate producing high levels of antigen [73]. Several recombinant vaccines have been developed against *Campylobacter* colonization in broilers. Neal-McKinney et al. [53] have reported that amongst several recombinant *Campylobacter* antigens tested, *FlaA, FlpA* and *a CaDF-FlaA-FlpA* fused protein resulted in up to 2 log_10_ reduction in *Campylobacter* colonization. Similarly, Theoret and colleagues [74] reported a 2.5 log_10_ reduction in *Campylobacter* colonization in chickens when a recombinant attenuated *Salmonella enterica* was used to synthesize a Dsp antigen. Wyszynska et al. [59] reported that oral vaccination with an avirulent *Salmonella* vaccine expressing *Campylobacter CjA* resulted in increased production of serum IgY and intestinal IgA, associated with up to 6 log_10_ reductions in cecal *Campylobacter* counts on day 12 post challenge. On the other hand, Buckley et al. [58] demonstrated that vaccination of chickens with *Salmonella* serovar Typhimurium expressing *CjaA* resulted in serum IgY and biliary IgA production, but only observed a 1.4 log_10_ reduction in cecal *Campylobacter*. Similar results were obtained in chickens vaccinated with an attenuated *Salmonella*-vectored *CjaA* protein on the day of hatch and challenged with *C. jejuni* after three weeks of vaccination [67]. Despite these promising results, however the need for two doses of vaccine at a two-week interval, followed by a 28-day withdrawal time (required for live *Salmonella* vaccines) before slaughter raises questions about the practicality of this approach. Moreover, one other challenge with *Salmonella* vectors is that they poorly colonize the chicken intestines, thus failing to prime the immune system effectively [75].

##### Nanoparticles-Based Vaccines

In recent years, various groups have investigated the potential of oral and *in-ovo* vaccination with nanoparticles-based vaccines. Recently, we demonstrated that vaccination of broiler chickens with *C. jejuni* lysate and a TLR21 ligand CpG ODN 2007, encapsulated in the poly (lactic-co-glycolic acid) (PLGA) nanoparticles, induced mucosal innate responses in the intestine and cecal tonsils, increased serum anti-*C. jejuni* IgY antibody (Ab) titers, modulated the composition of cecal microbiota and reduced cecal *C. jejuni* count by 2.4 log_10_ in vaccinated broiler chickens [62,76]. Huang and colleagues [60] tested the effect of intranasal immunization of chickens with chitosan nanoparticles containing a recombinant plasmid pCAGGS-*flaA* (a gene encoding a flagellin protein), on intestinal and cecal colonization with *Campylobacter*. The results revealed an increase in the serum IgY and intestinal secretory IgA with up to 3 and 2 log_10_ CFU/g reduction in *Campylobacter* loads in the large intestine and cecum, respectively. Despite it seems promising, however this vaccine was tested against only one strain of *Campylobacter* (*C. jejuni* ALM-80) and thus, further studies are required to assess its efficacy in a heterologous challenge model. On the other hand, Kobierecka et al. [61] reported that *in-ovo* immunization of 18-day-old embryonated chicken eggs with Gram-positive Enhancer Matrix (GEM) particles containing two *Campylobacter* antigens (*CjaA* and *CjaD*) reduced cecal colonization with *Campylobacter* by only 1 log_10_ in 3- and 4-week-old broiler chickens following challenge with a heterologous *C. jejuni* strain. In the same study, a higher reduction in *Campylobacter* count by 2 log_10_ was observed, when these antigens were encapsulated in a liposome. Although showing only a moderate level of *Campylobacter* reduction, these results suggest *in-ovo* immunization as a successful strategy that could potentially be enhanced with booster vaccinations post-hatching. Along similar lines, subcutaneous administration of a crude mixture of *C. jejuni* outer membrane proteins (OMPs)-loaded PLGA nanoparticles has been shown to induce systemic protective antibody responses and to reduce *C. jejuni* colonization below the limit of detection in broiler chickens [55]. However, the subcutaneous route is not deemed feasible for mass administration in poultry production.

Despite extensive research over the past few decades, none of the vaccines developed by different groups of researchers conferred “full protection” against infection with this bacterium in chickens [77]. Recent advancements in molecular approaches have opened new avenues for the identification of novel vaccine antigens, through strategies such as reverse vaccinology [78], which could be a steppingstone for vaccine preparation and optimization in the future.

#### 2.2.2. Feed Additives

##### Prebiotics

With efforts to minimize the use of antibiotics as growth promotants in poultry production, alternative strategies are urgently needed to compensate for their effects. Prebiotics are broadly defined as “indigestible fibers that beneficially influence the gut microbiome when used as feed additives”. Among these prebiotics, mannan-oligosaccharides, β-glucans, and fructans are the most used prebiotics in commercial poultry farming [79,80]. Mannan oligosaccharides (MOS) are largely derived from outer cell membranes of *Saccharomyces cerevisiae* yeast. These prebiotics are resistant to hydrolysis by digestive enzymes and are widely used in poultry feed to reduce pathogenic organisms in the gut and to enhance productivity. MOS is rich in mannoproteins, mannan, and glucan, and can inhibit gastrointestinal colonization of pathogens by binding to their type-1 fimbriae appendages and inhibiting lectin [81,82]. In general, supplementation of prebiotics alone does not appear to offer the best protection against *Campylobacter* in chickens. For instance, a recent study demonstrated that dietary supplementation of *Saccharomyces*-derived prebiotic reduced *Campylobacter* count by up to 1 log_10_ CFU per gram of cecal contents [83]. Similarly, Baurhoo and colleagues [84] have reported that dietary inclusion of 0.2% or 0.5% of MOS resulted in a minimal reduction in *Campylobacter* colonization by approximately 0.25 log. In addition to MOS, fermentation of long-chain fructans, which is extracted from plants, by gut microbes, results in the production of short-chain fatty acids (SCFAs) and lactic acid [85]. The abundance of these acids in the chicken gut promotes the growth and metabolic activity of beneficial microorganisms in the chicken gut and lowers luminal pH, thereby contributing to the prevention of pathogen colonization [85]. In this context, supplementation with chicory fructans, such as inulin or chicory oligofructose, significantly lowered cecum *Campylobacter* load by a 1.6 log_10_ [86]. Although several studies showed that dietary supplementation with single prebiotic results in a reduction in *Campylobacter* colonization, the simultaneous use of multiple prebiotics did not confer additional protection against *Campylobacter* infection. For instance, while a significant reduction in *Campylobacter* colonization was observed in chickens received fructans [86], co-administration of plum fibers, fructooligosaccharides, and galactooligosaccharides showed no such effect [87].

##### Probiotics and Their Products

Probiotics: Over the past few years, probiotics have received considerable attention as antimicrobial alternatives for in-feed antibiotics in the poultry diet [88,89]. Probiotics are defined as “beneficial live microorganisms that confer various health benefits to the host when used as supplements” [90]. In addition to their role in competing with microbial pathogens for adhesion and colonization sites and in modulating intestinal immune responses and microbiome composition [90,91,92], probiotic bacteria produce anti-microbial substances, such as bacteriocins, lactic acid, and hydrogen peroxide that possess direct bactericidal activity against enteric pathogens [88,89,90,93,94,95]. We have recently assessed the immunomodulatory and anti-*Campylobacter* activities of different *Lactobacillus* species, including *L. salivarius*, *L. johnsonii*, *L. reuteri*, *L. crispatus*, and *L. gasseri*, in vitro [96]. The results revealed that *Lactobacillus* species exhibited differential anti-*C. jejuni* activities as demonstrated by inhibition of *Campylobacter* growth, abrogation of the quorum sensing signal, inhibition of *Campylobacter* invasion in cultured intestinal epithelial cells and a reduction in the expression of *C. jejuni* virulence genes (except *L. reuteri*), including genes involved in motility (*flaA, flaB*, and *flhA*), autoinducer production (*luxS*), and invasion (*ciaB* and *iamA*). Additionally, *Lactobacillus* species have shown potential to enhance the phagocytic activity of chicken macrophages and modulate their immune responses as demonstrated by an enhanced expression of cytokines, including interferon (IFN)-γ, interleukin (IL)-1β, IL-12p40, and IL-10, chemokine, including CXCL8, and the co-stimulatory surface molecules, including CD40, CD80, and CD86 [96]. Similarly, probiotic *E. coli* strain Nissle 1917 (EcN) (free or Chitosan-alginate microencapsulated) has been shown to modulate the immune responses in intestinal cell lines [97,98,99].

Whilst many studies support the role of probiotics in providing protection against *Campylobacter* infection, the outcomes of these studies are greatly heterogeneous. It is unclear whether the inconsistencies in the effectiveness of probiotics are due to strain-specific effects and/or related to the differences of the age and type of the bird, dosage and combinations of probiotics, route of administration, dosing frequency, duration of application, and other environmental and managemental factors including the type of the housing and the dietary regimen (Table 2).

Various lactic acid-producing bacteria have been studied for their ability to reduce *Campylobacter* colonization in broiler chickens [119]. More specifically, the members of the genera *Lactobacillus* and *Bifidobacterium* are amongst the most widely used probiotics in the poultry industry [90]. Administration of *L. gasseri* to newly hatched chicks was shown to reduce cecal *C. jejuni* counts by approximately 250-fold at day 14 post infection [105]. Repeated oral administration of *L. salivarius*, every 2–3 days starting from day one to day 35 of age, has been reported to reduce cecal *Campylobacter* loads by 0.8 log_10_ at 14 days and a higher reduction of 2.8 log_10_ was observed at 35 days of age [102]. Recently, Helmy et al. [101] showed that oral treatment of chickens with free or chitosan-alginate microencapsulated probiotic *E. coli* Nissle 1917, three times per week for two consecutive weeks, reduced cecal *C. jejuni* colonization by 2 and 2.5 log CFU/g, respectively, and enhanced the growth performance, intestinal morphology and immunity of the treated chickens without adversely affecting the gut microbiota. While single species probiotics have been found to be effective in reducing *Campylobacter* colonization, combinations of several different probiotic species appear to be more effective [120]. For instance, a substantial reduction of up to 5 log_10_ CFU/mL in cecal *Campylobacter* count has been reported with the use of probiotic mixtures containing *L. paracasei* J.R and *L. rhamnosus* 15b, *L. lactis* Y and *L. lactis* [120]. Ghareeb et al. [119] have also reported a reduction in the cecal load of *C. jejuni* by up to 6 log_10_ in broiler chickens received a mixture of probiotics containing *Bifidobacterium animalis*, *Enterococcus faecium*, *L. salivarius*, and *L. reuteri*, and *Pediococcus acidilactici*. 

By-products of probiotic bacteria: Bacteriocins are ribosomally synthesized peptides that possess antimicrobial activity against other bacterial strains which may or may not be taxonomically related [123,124]. Indeed, several studies reported a substantial reduction in *Campylobacter* colonization following the administration of various bacteriocins. For example, Messaoudi and colleagues [125] reported a 2 log_10_ reduction in *Campylobacter* count following in vitro exposure to bacteriocins derived from *L. salvaris* SMZD51. Similarly, Stern et al. [126] have reported that administration of bacteriocin OR-7 derived from *L. salivarus* NRRL B-30514 can reduce *Campylobacter* colonization by up to 6 log_10_ units. Furthermore, Stern et al. have reported that administration of a class 2a bacteriocin, derived from *Paenibacillus polymyxa* NRRL B-30509, resulted in significant reductions in *Campylobacter* colonization, to the extent that at certain time points of the trial, *Campylobacter* was reduced to undetectable levels, while an average of 7.2 log_10_ CFU/g of *Campylobacter* was detected in the non-treated, infected birds. In another study conducted on turkey poults, co-administration of bacteriocin OR-7 and bacteriocin B602, derived from *Paenibacillus polymyxa* NRRL B-30509, reduced *C. coli* colonization to below detectable levels in the duodenum and cecum [127].

It is noteworthy that, while administration of live bacteriocin-producing bacteria (*L. salivarius* NRRL B-30514 and *Paenibacillus polymyxa* NRRL B-30509) failed to protect against *Campylobacter*, their bacteriocins were shown to reduce colonization by up to 6 log_10_ [128]. However, although various bacteriocins have shown potential to reduce the *Campylobacter* burden in chickens (Table 3), the development of bacteriocin resistance and their impact on the gut microbiota need to be investigated further.

##### Synbiotics

To increase the efficacy of probiotics, studies have found that co-administration of prebiotics provides an additive or synergistic protection against *Campylobacter* infection [133,134]. For example, although not observed in all clinical trials, administration of *Bacillus* spp., *L. salivarius* subsp. *salivarius* and *L. salivarius* subsp. *salicinus* alone has been shown to lower *Campylobacter* counts by 1–2 log_10_, while when combined with 0.4% MOS, *L. salivarius* subsp. *salicinus* resulted in a 3 log_10_ reduction in *Campylobacter* counts [116]. Lifelong dietary supplementation of synbiotic (probiotic strain *Bifidobacterium longum* subsp. longum PCB13 and prebiotic Xylooligosaccharides) to chickens challenged with *C. jejuni* strain M1 showed a better efficacy compared to short-term supplementation [135]. Together, these results indicate that concurrent administration of prebiotics with probiotics augments their potential to combat *Campylobacter* colonization.

##### Essential Oils

As the demand for antibiotic-free poultry products grows, several alternative strategies, such as essential oils (EOs), are being identified at pre- and post-harvest stages [136]. Essential oils are “volatile or ethereal oils with oily plant-based liquids that possess aromatic properties” [137]. They are extracted from plants using hydrodistillation, steam distillation, or solvent extraction to produce concentrate of aromatic and volatile compounds [137]. A series of studies conducted by Solis de los Santos et al. [138,139] have demonstrated that providing a 0.7% caprylic acid, extracted from coconut oil and palm kernel oil, in feed at concentrations below 1% significantly reduced *Campylobacter* colonization in the cecum of broilers at different ages. The effectiveness of plant extracts, containing natural essential oils, against *Campylobacter* colonization in chickens has also been assessed [140]. Although a minimal reduction in *Campylobacter* colonization was observed following dietary inclusion of thymol and carvacrol by 2% and 1%, respectively, their combination did not have additive effects as demonstrated by only 0.5% reduction in *Campylobacter* counts following their supplementation compared to either one alone. Furthermore, a mixture of garlic and cinnamon extract, which are rich sources of essential oils, reduced *Campylobacter* colonization in the cecum by 1 log_10_ CFU/g at 3 days post-infection (day 11 of age), whereas no significant effects were observed on day 35 or day 42 of age [140]. In the same study, no additive effects were observed when a combination of these plant extracts with other additives, including prebiotics and other herbs, were supplemented to chickens.

Recently, Szott et al. [141] showed that adding carvacrol in broilers feed (at a concentration of 120 mg/kg feed) for 4 days reduced *C. jejuni* colonization by up to 1.2 log_10_ in the cloacal swabs and colon between 1 and 28 days of age, while no significant effect was observed on cecal colonization at 33-day-old of age. In another study [142], feeding 0.3% trans-cinnamaldehyde-coated feed to *C. jejuni*-infected broilers exhibited no significant reduction in *Campylobacter* colonization in the cecum after 1 week. In an in vitro model, treatment of chicken cecal contents with different concentrations (10, 20, and 30 mM) of thymol, eugenol, and carvacrol and then spiking them with 10^5^ CFU/mL *C. jejuni* was found to reduce *C. jejuni* to undetectable levels after 8 h of incubation, while treatment with trans-cinnamaldehyde reduced the levels to less than 1 log_10_ CFU/mL at the same incubation time [143]. In an ex vivo study, Kurekci et al. [144] used a fermentation assay to evaluate the anti-*Campylobacter* activity of three essential oils, including tea tree oil, lemon myrtle oil and Leptospermum oil. Addition of these oils to cecal content of 20-day old chicken spiked with 3 × 10^8^ CFU/mL of *C. jejuni* reduced *C. jejuni* concentrations by 3.3 log_10_ CFU/mL, without altering the fermentation profile of the cecal microbiota. Collectively, these findings suggest the use of these essential oils to reduce *Campylobacter* burden in poultry; however, dose optimization and different treatment modalities may be required to attain more desirable outcomes.

##### Organic Acids

Organic acids (OAs) are naturally occurring organic compounds that retain acidic properties and are distinguished from other acids by the functional group -COOH [145]. OAs are mainly composed of SCFAs (≤C6), such as formic, propionic, acetic, lactic, butyric, and other medium-chain (C7 to C10), and long-chain fatty acids (LCFA; ≥C11) [145,146]. Previous studies suggested that OAs can be used as acidifiers in poultry drinking water and as antimicrobial feed additives [145]. In addition to their antimicrobial activities, dietary inclusion of OAs has been shown to increase feed conversion efficiency, nutrient digestibility, and to modulate anti-oxidative status of the gastrointestinal tract of chickens [145,146,147]. However, the exact mechanisms of action of these OAs remain unclear. Dittoe and colleagues [148] reported that since compounds comprising organic acids are acidic, dietary supplementation of these compounds to chickens alters the pH of the gastrointestinal tract and thus, shielding it from pH-sensitive pathogens. However, in another study, an association was observed between the concentration of dissociated organic acids, but not the pH, and inhibition of *C. jejuni* growth following exposure to various organic acids. Regardless of the discrepancies on the mechanisms of action of OAs, these findings suggest the on-farm use of these OAs to control bacterial pathogens [149].

In fact, the use of combinations of various organic acids was shown to produce additive or synergistic effects. For instance, Peh et al. [150] reported that a combination of caprylic acid, sorbic acid and caproic acid exhibited synergistic effects on six *C. jejuni* and four *C. coli* isolates and reduced the MIC90 values of these compounds using broth microdilution method. Despite evidence indicating that OAs possess potent bactericidal activities when tested in vitro, it should be noted that the OAs may not exhibit the same activity when used in live birds. This is consistent with the findings of Hermans et al. [151] who also demonstrated that low concentrations of caprylic or capric acids (4 mM) and caproic acid (16 mM) killed six *C. jejuni* strains within 24 h when tested in vitro; however, these marked bactericidal effects did not replicate when tested in vivo. Nonetheless, other studies have pointed out that a more desirable outcome could be achieved by optimization of organic acid concentration. For instance, while a combination of 1% formic acid and 0.1% sorbate did not reduce cecal *Campylobacter* counts, a higher concentration 1.5–2% of formic acid and 0.1% sorbate resulted in complete elimination of *C. jejuni* colonization in chickens [152]. In another study, a dose-dependent reduction in *Campylobacter* count was observed following dietary supplementation of propionic, sorbic acids and pure botanicals [153]. In addition to their synergistic effects when used in a combination, the microencapsulation of these acids was shown to enhance their bactericidal activity as demonstrated by a significant reduction in *Campylobacter* counts by up to 5.2 log_10_ at 42 days of age. Nonetheless, this study did not assess the effects of these organic acids on feed intake and growth performance of chickens. Although their undeniable beneficial effects, it is important to note that inclusion of these organic acids in poultry feed may affect its palatability [154], thereby reducing chicken feed intake. Regardless of these potential limitations, the effectiveness of organic acids appears to be largely dependent on the type, concentration, and combination of the organic acid used.

##### Small Molecule Inhibitors

Small molecule (SM) is defined as “a low molecular weight organic compound, involved in molecular pathways by targeting important proteins” [155]. SM inhibitors are promising alternatives to antibiotics that can be utilized for the control of *Campylobacter* infection in poultry. They possess the ability to target specific pathways in bacterial cellular processes and perform narrow-broad spectrum antimicrobial activity [156]. In addition, SMs have a long half-life which broadens their potential in the clinical applications of the drug. Although SMs can be effective alternatives to antibiotics, they have their own sets of pharmacological limitations: foremost among these limitations is that the small and compact structure required for the broader bioavailability of the drug decreases its specificity and sensitivity over time [157]. Johnson and his group have screened a pool of 147,000 SMs inhibitors, with several compounds have shown inhibitory activity against *C. jejuni.* These compounds have demonstrated remarkable ability to suppress motility and biofilm formation without exerting cytotoxic effects on eukaryotic cells. The inhibitory activities of these compounds were also tested in one-day-old chicken experimentally infected with *C. jejuni* and the results revealed that only campynexin A was able to decrease cecal colonization by 1 log [158].

In a similar study, Deblais et al. [159] screened a library of about 4200 SMs and identified SM based on thiophene sulfonamide with activity against *C. jejuni* 86–176 and *C. coli* ATCC33559. The results revealed a reduction in cecal *Campylobacter* load of three-week-old chicken by 1 log and 2 log following treatment with a benzyl thiophene sulfonamide based small molecule compounds (TH-4 and TH-8), respectively. Additionally, while no significant changes in the microbiota were observed in the TH-8-treated chickens, TH-4 treatment increased the abundance of *Coprococcus* by 2.57-fold, and a reduction in *Peptostreptococcacae, Erypelotrichacae* cc115, and *Eubacterium* abundance by 5.52-fold, 9.6-fold, and 3.0-fold, respectively.

Among 4182 bioactive SMs compounds screened against *C. jejuni* by Kumar et al. [160], only 478 had a bactericidal effect and 303 had a bacteriostatic effect. A further screening of 79 bactericidal compounds was conducted against different *C. jejuni* isolates, with only 12 compounds showed consistent bactericidal effect on *C. jejuni*. These 12 compounds had shown minimal cytotoxic effect on Caco-2 cells and hemolytic effect on sheep RBC. Following further screening, 10 compounds completely cleared *C. jejuni* populations even at the concentration of 25 µM under in vitro conditions.

Evidence indicates that plant-based compounds can also be used as anti-*Campylobacter* therapeutics. It was reported that phenolic compounds exhibit antimicrobial activity against both antibiotic-sensitive and -resistant strains of *Campylobacter*. Among 9 phenolic compounds used, epigallocatechin gallate (EGCG) and carnosic acid had shown a significant anti-*Campylobacter* effect with a MIC of 78 µg/mL and 19.5 µg/mL, respectively. Similarly, rosemarinic acid showed antagonistic activity against *C. jejuni* with a MIC of 158 µg/mL [161]. Notably, most of the studies evaluating the antimicrobial activity of various SMs have been carried out in vitro with only a few studies have reported their *efficacy* in vivo. Further in vivo research is strongly needed to validate the in vitro observations and determine whether SMs could substitute in-feed antibiotic and combat *Campylobacter* in poultry.

##### Short Chain Fatty Acids

Short chain fatty acids (SCFAs), including acetate, propionate, butyrate, iso-butyrate, valerate, iso-valerate, hexanoate, are microbial metabolites produced by fatty acid-producing bacteria, such as members of phyla *Bacteroides* and *Firmicutes* in the gut [162,163,164]. In addition to exhibiting immunostimulatory and bactericidal properties, the abundance of these metabolites lowers the pH of the gut, making the condition unfavorable for the growth of pathogenic bacteria that are sensitive to acidic pH [163]. For instance, van der Wielen et al. [165] have observed a significant reduction in *Enterobacteriaceae* growth following in vitro exposure to volatile fatty acids and when administered to chickens, undissociated acetate, propionate, and butyrate was shown to reduce *Enterobacteriaceae* levels in the gut. However, although butyrate was shown to exhibit bactericidal activity against *Campylobacter* in vitro, the use of butyrate-coated micro-beads as a feed additive did not reduce *C. jejuni* colonization in the cecum of two-week-old broiler chicks [107]. The authors attributed this to a rapid absorption of butyrate by the enterocytes and speculated that elevation of butyrate level could probably result in a decrease in *C. jejuni* colonization. It is therefore conceivable that administration of SCFAs-producing bacteria may potentially provide a continuous source of butyrate, which may, in turn, reduce *Campylobacter* colonization in broiler chickens.

##### Bacteriophages

Bacteriophages are “viruses that can infect and kill targeted bacterial cells” [166]. They have been extensively researched and used as antimicrobial agents worldwide for the treatment of several human diseases and recently as preventative and therapeutic agents to control *Campylobacter* colonization in poultry [167,168,169,170,171,172]. In a recent study, oral administration of two field bacteriophages to experimentally infected broiler chickens at 37 days of age significantly reduced *Campylobacter* counts by 2 log_10_ CFU/g at 40 days of age [168]. Fischer et al. [169] have also demonstrated a significant reduction in cecal *Campylobacter* count by up to 2.8 log_10_ CFU/g following administration of a single NCTC 12673 or multiple phage cocktail consisting of phages NCTC 12672 12673 12674 and 12678. Interestingly, while NCTC 12673 alone showed similar efficacy to the phage cocktail, administering the phage cocktail significantly reduced initial resistance levels, suggesting that multiple phages may result in delaying the onset of *Campylobacter* infection. Similar observations were made by Kittler et al. [170] who also found that administration of the same bacteriophage cocktail results in a significant reduction in fecal shedding of *Campylobacter* by approximately 3 log_10_ in chickens at the slaughter age. Additionally, in this trial *Campylobacter* count was reduced below the detection limit during the first 24 h of its administration. Together, these findings suggest that a combination of selected phages administered a few days prior to slaughter may substantially reduce *Campylobacter* levels in poultry meat.

In the context of their therapeutic efficacy against *Campylobacter*, a previous study showed that administration of NCTC12671 and 12669 at 10^10^ PFU to 39-day-old chickens after 7 days of challenge with *Campylobacter* resulted in a significant reduction in *C. jejuni* by up to 1.5 log_10_ [167]. A higher, but transient, reduction in *Campylobacter* colonization was observed when phage 71 was administered for ten consecutive days to younger birds (two-week-old chicks) by up to 3 log_10_, suggesting the need for a booster dose or continuous administration to maintain colonization resistance against *Campylobacter*. In view of this, previous studies have shown that administering phages during the early stages of chicken’s life, followed by a booster dose at the end of the production cycle results in a greater reduction in *Campylobacter* colonization. For example, oral administration of phage 71 to 7-day-old chicks, followed by *Campylobacter* challenge on day 10 of age, and phage treatment until day 16 of age, resulted in a delay in the onset of *Campylobacter* colonization as well as a reduction in cecal *C. jejuni* count by 1 log_10_ [167].

In addition to possible synergistic interactions, using multiple phages equipped with various defense mechanisms is thought to result in optimal efficacy due to the possible development of phage resistance when used alone, which is likely the reason for the transient reduction in *Campylobacter* levels observed immediately after phage administration followed by stabilization, and rebounding of *Campylobacter* levels. However, phage cocktails must be carefully designed based on the types of phages being administered to achieve maximal benefits as phages exhibit different mechanisms of infection. For example, group 2 phages isolated from *C. jejuni* RM1221 typically use flagella as a route of entry, while group 3 phages isolated from *C. jejuni* NCTC12662 target bacterial capsular polysaccharide receptors [171,173]. In this context, Hammer et al. [174] have demonstrated that administration of a group 3 phage CP14 alone reduced fecal counts by 1 log_10_, while when co-administered with CP81, which belongs to the same group, no further reduction in fecal *Campylobacter* count was observed. On the other hand, a greater reduction > 3 log_10_ of fecal *Campylobacter* counts was observed in three-week-old chickens when a group 2 CP61 phage was administered 24 h following administration of group 3 CP14 [174]. These findings indicate that concomitant administration of phages belonging to the same family may result in greater phage resistance and warrant proper selection of phages from different groups to achieve optimal efficacy.

Despite the number of studies reporting significant efficacy in experimental settings, the major shortcomings of commercial application of bacteriophages, include the high specificity of the selected phages, as such universal applicability and efficacy of phage strains are not achievable, in addition to the lack of evidence over their safety and stability [175]. For example, in a previous study, phage CP8 exhibited variable effectiveness against different serotypes of *C. jejuni* [172]. In chickens experimentally infected with either *C. jejuni* GIIC8 or HPC5 serotypes, the efficacy of orally administered CP8 to 25-day-old chickens was notably higher in GIIC8 infected birds with substantial reductions in *Campylobacter* in the cecum by up to 5.6 log_10_ within 24 h of phage administration, and final reductions of up to 2.1 log_10_, while CP8 administration in HPC5 infected birds resulted in no significant reduction. In the same study, phage CP34 exhibited an opposite pattern of efficacy between the two *Campylobacter* serotypes with significantly greater reductions seen in HPC5 infected chickens of up to 3.9 log_10_ within 24 h of phage administration [172]. Overall, although bacteriophages have shown remarkable reduction in *Campylobacter* colonization, previous findings highlight the critical shortcoming of bacteriophages being highly specific for their targets. As such, challenges will be faced in finding suitable phages, which can be effective against a large number of *Campylobacter* serotypes, to be widely used in the poultry industry.

#### 2.2.3. Fecal Microbial Transplant and Microbial Consortia

The protective role of the gut microbiota against *Campylobacter* colonization has been recently investigated. Higher levels of *Campylobacter* were observed in the cecal contents, spleen, liver, and ileum of chickens experimentally infected with *Campylobacter* and raised under germ-free conditions and also in chickens with a compromised microbiome than in conventionally farmed chickens [176]. Furthermore, *Campylobacter* infection was shown to result in the development of intestinal lesions in these groups implying that *C. jejuni* may not simply be a commensal microbe in the poultry gut, but rather a pathogen that has symptomatic effects, notably in hosts with atypical gut microbiome [177].

Considering the integral role of the gut microbiome in the intestinal immune system development and defense against pathogens [178,179], there has been an increasing interest in manipulating the gut microbiome for prevention of enteric infections and colonization by food-borne pathogens, including *C. jejuni*. While the prophylactic use of antimicrobial alternatives has been studied for their potential to induce anti-*Campylobacter* mucosal immune responses, and although such strategies have independent mechanisms of action, their indirect effects on the microbiome may have significant contributions to the observed efficacy. Recently, we have reported changes in microbiome compositions in chickens administered with PLGA-encapsulated CpG oligodeoxynucleotides and *C. jejuni* lysates [62]. In addition to the lower numbers of *C. jejuni* observed in the treated chickens, significantly higher levels of microbial diversity, particularly members of phyla *Firmicutes* and *Bacteroidetes*, were also observed.

Manipulation of the gut microbiota via probiotics and fecal microbial transplants (FMT), particularly at early points in the production cycle, has shown significant potential to reduce *Campylobacter* colonization. At present, a very limited number of studies have investigated the ability of gut microbiome transplants to reduce *Campylobacter* colonization in broilers. Gilroy et al. [180] have recently reported undefined fecal transplants obtained from eight-weeks old chickens, which were free of *Campylobacter*, resulted in lower levels of colonization in chicks that were exposed to *Campylobacter* either through a direct challenge or a seeder bird model. Furthermore, changes in the microbiome were observed with FMT-treated birds having greater phylogenetic diversity amongst the species constituting the gut microbiome [181]. A significant increase in the levels of lactobacilli was observed in chicks received FMTs, with an average of a 4.5-fold increase in the abundance of *Lactobacillales* in FMT birds relative to controls. Conversely, a 1.78-fold decrease in the abundance of the order *Clostridiales*, which was relatively abundant in the gut microbiome of chickens colonized by *Campylobacter* [180]. However, it should be noted that FMT lacks batch-to-batch consistency and microbial identification is needed to avoid potential carryover of undesirable microorganisms from apparently healthy chickens to recipient chickens. Research efforts have been undertaken to overcome these challenges by developing a well-characterized competitive exclusion culture (microbial consortium) which constitutes a safer alternative to the FMT [182]. Besides their role in improving gut health, supplementation of a consortium of beneficial bacteria to newly hatched chicks allows early colonization of the gastrointestinal tract and may preclude pathogen’s attachment to the intestinal mucosal surface. We have recently demonstrated that addition of competitive exclusion cultures (Aviguard and CEL) to drinking water immediately post-hatch resulted in a significant reduction in *Campylobacter* colonization in broiler chickens relative to treatment with bacitracin and a lower, but non-significant, reduction relative to the untreated controls, albeit, these treatments resulted in an increase in the relative abundance of *Bacteroidaceae* and *Rikenellaceae*, with both playing a role in improving host immunity and metabolic functions [182]. Aside from their minimal role in reducing *Campylobacter* colonization, the ability of these consortia to modulate the microbiome composition, during the course of *Campylobacter* infection, suggest their potential use as a safer alternative to bacitracin in poultry feed to tackle the growing threat of antibiotic resistance.

## 3. Post-Harvest Control Measures (Production Chain Interventions)

On-farm control measures alone have not been sufficient to eliminate *Campylobacter* in poultry. Sanitation practices in poultry processing facilities should also be implemented to further reduce *Campylobacter* levels at later stages of the food supply chain. In a recent European Food Safety Authority (EFSA) report [183], the proportion of broiler flocks infected by *Campylobacter* varies widely (ranging from 2 to 100%), and strongly correlates with the prevalence of *Campylobacter* on broiler carcasses (4.9% to 100%). Chickens carry a high load of *Campylobacter* of approximately 8 log_10_ CFU/g in their caeca prior to slaughter. Contamination of chickens’ feathers with fecal material during transportation to the slaughterhouse can also be a significant external source of carcass contamination during the plucking/defeathering process [184,185].

Transportation crates are a particularly problematic source of cross-facility transmission of *Campylobacter*, as birds are kept in crates for extended periods of up to 3–12 h, such as before transportation or slaughter, during which the crates are soiled with fecal droppings and *Campylobacter* shedding can be especially high [25]. Even with measures taken for cleaning and disinfecting transportation crates, *Campylobacter* was detected in 57% of the swab samples collected from cleaned crates and a notable increase in the number of infected chickens by 9% was also observed in the cloacal swabs following transportation, but whether the increase in the number of infected birds enhanced the risk of carcass contamination during processing was not investigated in this study. 

In addition to possible stress-induced impairment of the gastrointestinal tract, feed restrictions during these times may also result in increasingly neutral pH levels in the gastrointestinal tract, providing a microenvironment suitable for optimal *Campylobacter* growth [25,26]. Increased turnaround periods have also been reported to reduce the risk of new flocks becoming colonized. This is largely because *Campylobacter* is increasingly less effective at colonization with greater periods spent outside the host gastrointestinal tract. Lazaro et al. [186] have reported that *Campylobacter* can survive up to 7 months in a viable but not culturable state. As such, increasing down time of crates between flocks and effective cleaning could put in place to lower the risk of horizontal exposure. In addition to these measures, effective carcass decontamination practices should be considered to reduce *Campylobacter* concentration in the poultry meat. Contamination of meat products by gut contents is difficult to prevent during processing at slaughter plants because of the high numbers of *C. jejuni* in the gut, and the large percentage of birds infected. Data from surveillance studies in different countries indicated a high prevalence of *Campylobacter* on raw retail chicken meat. For example, *Campylobacter* was detected in 28.6%, 36.5%, 41.2% and 52.2%, 59.9% of samples from chicken meat from retail stores in the United Arab Emirates, Qatar, the United Kingdom, Saudi Arabia, and Canada, respectively [187,188,189,190,191]. It is estimated that a reduction in *Campylobacter* counts in the neck and breast skin to 10^3^ CFU/g reduces the public health risk by 50% [192]. The following section summarizes the physical, chemical, and biological control measures that have been, or are being taken, to reduce *Campylobacter* load on poultry carcasses.

### 3.1. Slaughter Plants Cleaning and Sanitation

In addition to the risk of carcass contamination during evisceration, cross contamination from contaminated processing equipment surfaces, due to insufficient cleaning and disinfection, should also be considered as another source of carcass contamination during the slaughter process. Sounmet and Sanders were the first to report the survival of *Campylobacter* on cleaned and disinfected surfaces of four French slaughterhouses [193]. It is, however, unclear, whether the transportation crates or the previously slaughter flock was responsible for the observed contamination, since the same *Campylobacter* strain was isolated from the crates and chicken carcasses. These data highlight the importance of carcass treatment at the end of the process line.

### 3.2. Carcass Decontamination

Several interventions used at the slaughterhouse level have been assessed for their degree of impact on reducing human campylobacteriosis and have extensively been reported elsewhere [194]. Amongst some of the greatest potential risk reductions, the use of carcass decontamination methods, including physical, chemical, and biological methods, is of notable potential because they can be fully implemented with the introduction of mandatory government food safety regulations at the later stage of the food chain.

#### 3.2.1. Physical and Chemical Methods

Poultry carcass treatment with 2 % lactic acid is estimated to reduce the risk of *Campylobacter* infection in humans between 37–56 %, whereas treatment with acidified sodium chlorite (1200 mg/L) and trisodium phosphate (10–12 %, pH 12) was found to reduce the risk by 75–96%, and 67–84%, respectively [194]. In the US, organic acids and quaternary ammonium compounds have also been used for decontamination at the slaughterhouses [195]. Slaughterhouse stage is also an optimal point in the production cycle for decontamination with irradiation and/or cooking. U.V light and high temperatures have long been used to reduce meat contamination and have the potential for very high levels of efficacy if they are successfully implemented in high volume slaughter lines [196]. Overall, despite the large degree of efficacy for these various carcass decontamination methods, a major drawback of the currently available methods is the effect they have on the sensory attributes of the meat. In particular, freeze–thaw cycles, irradiation and precooked meats are amongst the most unfavorable qualities for consumers concerning sensory attributes and overall perception of the food [197].

In addition to different decontamination methods, poultry meat is often frozen during some point in the farm to fork continuum. Although not offering a significant reduction in *Campylobacter* counts to be regarded as a primary solution to reduce *Campylobacter* at the production level, freezing can result in moderate reductions in the pathogen on poultry meat. Despite this, it was reported that freeze cycles have low efficacy in reducing *Campylobacter* levels on poultry carcasses in contrast to some of the previously mentioned compounds, such as sodium triphosphate sprays [26]. Nonetheless, a variety of factors such as duration of freeze periods, numbers of freeze–thaw cycles as well as methods of freezing may contribute to the discrepancies within the literature [198]. Thermophilic *Campylobacter* growth occurs between 37–42 °C, while retardation of growth occurs below 30 degrees [199]. Although *Campylobacter* is unable to grow below 30 degrees, it can survive in temperatures as low as 4 °C under moist conditions for periods of up to 7 months [199,200,201]. Bhaduri et al. [199] have reported reductions in *Campylobacter* counts between 0.31 to 0.81 log_10_ CFU/g after refrigeration at 4 °C for 3–7 days. While freezing *Campylobacter* is known to kill the bacteria to some extent, each freeze cycle is only able to eliminate a portion of the total bacterial load. As such, Sampers et al. [201] have reported that freezing poultry to −22 °C is effective at reducing the pathogen by about 1 log_10_ units, over the initial 24 h period, however beyond that the bacterial load remains relatively stable. Similarly, Bhaduri et al. [199] have reported that freezing to −20 °C results in *Campylobacter* reductions between 0.56–3.39 CFU/g over two weeks.

Various models have been applied to estimate the translational efficacy of reducing *Campylobacter* levels in live chickens and poultry meats and the downstream impact these reductions have on the incidence of human campylobacteriosis. Estimated risk reductions for human cases of campylobacteriosis from freezing broiler carcasses vary based on the length of the freeze cycle, with short time freezes of 2–3 days resulting in an estimated 62–93% risk reduction, while longer freeze cycles of three weeks are estimated to result in an 87–98% risk reduction [201]. Similar degrees of risk reduction are seen with hot water immersion and irradiation/cooking, up to 75–89% and 100%, respectively [192]. Furthermore, at the poultry farm level, disease risk reduction models have been used to estimate to what extent certain levels of reductions in chicken cecum *Campylobacter* colonization can contribute to reducing human campylobacteriosis [192]. In a most recent modelling approach, cecum concentration reductions of up to 2 or 3 log_10_ units are estimated to reduce human disease by 42% or 58%, respectively [183].

Although ultrasonication technology has been proven safe and effective for water purification and decontamination of carcasses and meat surfaces, numerous studies indicated that its application to *Campylobacter*-contaminated chicken meat and skin did not significantly reduce *Campylobacter* numbers. Detailed information on the effectiveness of other physical and chemical methods to reduce the concentration of *Campylobacter* on chicken carcasses during the slaughter process have been reviewed elsewhere [192,202].

#### 3.2.2. Biological Methods

As consumer demands for the availability of high quality and safe food products increase, *C. jejuni* decontamination and preservation using natural and chemical-free mechanisms has received greater attention by the industry. Some of the biological intervention technologies (BITs) that have shown promise in reducing food-borne pathogen load post-harvest include EOs, bacteriophages, bacteriocins, and probiotics. The effects of these strategies on pre-harvest control of *C. jejuni* have been described above. While research on the effect of BITs to reduce *C. jejuni* load pos-harvest is sparce, findings on other pathogens of food safety concern, such as *Salmonella* spp. and *Listeria* spp. indicates their potential for *C. jejuni* control post-harvest [203,204].

Probiotics and bacteriocins: two lactobacilli (*L. salivarius* and *L. hamsteri*) and 1% and 2% caprylic acid, alone or in combination with 2% chitosan solution, resulted in a consistent reduction in the number of *C. jejuni* on chicken wingettes that lasted for at least 5 days post treatment [205]. In another experiment, *B. longum* ssp. longum PCB133 resulted in 1.16 log CFU/g reduction in *C. jejuni* from chicken legs inoculated with 5.30 log CFU/g *C. jejuni* and packaged under a modified atmosphere (50% CO_2_/10% O_2_/40%N_2_) [206]. Previously, treatment of chicken meat and skin with 500 IU/gm nisin, a bacteriocin produced by *L. lactis*, in combination with 2% (*w*/*w*) sodium lactate, resulted in a significant reduction in *Arcobacter butzlerei*, a common *Campylobacter*-like organism with clinical and microbial features similar to *C. jejuni*, by up to 1 log [207].

Bacteriophages: In addition to their role in reducing *C. jejuni* counts in poultry pre-harvest, bacteriophages have also been tested with effective reduction in *C. jejuni* post-harvest. Application of a single 10^7^ PFU of group III phage φ2 (NCTC 12674, ACTC 35922-B2) therapy to chicken skin, inoculated with 10^6^ CFU of *C. jejuni* and stored at 4 °C for 10 days, resulted in 1 log CFU/cm^2^ reduction in this bacterium, and with further reduction of 2.5 log CFU/cm^2^ during an additional storage of poultry skin at −20 °C [208]. A recent study also showed the potential of bacteriophages against *C. jejuni* on chicken skin with a 0.73 log 10 reduction in the bacterial counts [209]. Another study by Zmapara et al. [210] showed that the application of Innolysins, which combine the enzymatic activity of endolysins with the binding capacity of phage receptor binding proteins to enhance endolysins activity against Gram-negative bacteria, on chicken skin refrigerated to 5 °C and contaminated with *C. jejuni*, resulted in 1.18 to 1.63 log reduction in this bacterium.

Essential oils: In addition to their potential application for pre-harvest control of *Campylobacter*, EOs have also shown a promising effect on the control of *C. jejuni* post-harvest. Djenane et al. [211] showed a 5 log CFU/g reduction in *C. jejuni* in skinless chicken breasts stored in microaerobic condition at 3 ± 2 °C, treated with Inula graveolens (MIC of 0.2%), Laurus nobilis (MIC of 0.6%), Pistacia lentiscus (MIC of 0.6%) and Atureja gontana (MIC of 0.6%) and experimentally contaminated with 5 × 10^5^ CFU/g of *C. jejuni* compared to untreated but contaminated control, which reached about 8 log 10 CFU/g after 1 week. Shrestha and colleagues [212] have also recently demonstrated that washing chicken carcasses with carvacrol significantly reduced *C. jejuni* on chicken skin by approximately 2.4 to 4 log_10_ CFU/sample. In addition to the effect on poultry products, essential oils can also play a role in reducing biofilm formation at the processing plant, which is challenging to control using commonly used antimicrobials and sanitizers, such as chlorine and peracetic acid. The biofilm formation of *C. jejuni* and *C. coli* were reduced by 70 to 80% with the treatment of coriander and linalool at concentrations of 2 μg/mL, and a lower concentration of up to 0.025 μg/mL resulted in reduction in biofilm by 10 to 20% compared to untreated biofilm controls [213]. While the above BITs show potential in reducing *C. jejuni* load post-harvest, more research is needed to develop effective formulations that can replace chemical intervention strategies.

Altogether, although these treatments appear to be promising in treatment of chicken skin and meat; however, their use in slaughter and processing plants has not yet been implemented, probably due to regulatory hurdles.

### 3.3. Eggshell Sanitation

Transmission of *Campylobacter* from fertile eggs to commercial flocks: Some of the initial studies conducted in the 1980s to investigate the ability of *Campylobacter* for shell penetration, reported the phenomenon to rarely occur and in a temperature-dependent manner [214,215]. *Campylobacter* can penetrate eggshell at 4 °C, with the viability once inside the eggs generally reported to be less than 72 h [214,215]. Although vertical transmission from laying hens to the progeny has rarely been reported to occur [37], it is unclear to what degree *Campylobacter* presence in the reproductive tract of hens may be contributing to infertility and inviable offspring. To date, there is no clear evidence that *Campylobacter* can be vertically transmitted from parent breeders to fertile eggs. It is also remains unclear whether trans-shell penetration by *Campylobacter* from the external environment poses a significant risk factor to commercial flocks [183]. Overall, the literature suggests that strategies that control *Campylobacter* transmission to flocks need to focus on routes other than vertical transmission through eggs.

Transmission of *Campylobacter* from table eggs to humans: *Campylobacter* was reported to be viable in the egg yolk stored at 18 °C for up to 14 days, and in the albumen and air sac for up to 8 days [216]. However, the authors also reported that in realistic settings although between 4–6% of newly laid eggs were *Campylobacter* positive, after storage of the eggs at 18 °C for 7 days, no viable *Campylobacter* remained. This raises the possibility that consumption of freshly laid eggs in a non-intensive production system could pose a significant threat to human health.

Despite the ability to penetrate the shell and survive in the yolk, numerous studies have shown that *Campylobacter* is extremely sensitive to atmospheric conditions. For example, Neill et al. [215] reported that these bacteria were unable to survive for more than 6 h when present in eggs that were incubated at 37 °C and exposed to a ventilated atmosphere. Yet, previous studies have reported a small number of viable organisms are recovered from the contents of chicken eggs. While most of the literature has focused on strategies in reducing *Campylobacter* presence on poultry meats, relatively little has been done to reduce *Campylobacter* transmission from eggs.

Altogether, the literature suggests that the combination of industry setting exposures such as fumigation, storage at cool temperatures and chemical sanitation such as quaternary ammonium, sodium hydroxide, phenol formaldehyde and hydrogen peroxide, together can significantly prevent *Campylobacter* survival. However, the concerns over the potential presence of chemical residues on the eggshell suggests the need for safer alternatives to improve microbiological safety and reduce the presence of chemical hazards in eggs. Physical sanitation of eggshell, such as ultraviolet light, infrared and ozone, has been proposed in some studies but have not yet been implemented [217,218].

## 4. Conclusions and Future Prospects

The continued increase in the incidence of human campylobacteriosis, which is estimated to be increased by 70% in 2018 from 2006 data, and associated healthcare costs necessitate an urgent need for effective ways to combat *Campylobacter* infection in poultry and prevent its transmission to humans through contaminated poultry products. Although there is no effective intervention measure available to “completely” eliminate *Campylobacter* in poultry, there is a considerable amount of promise for the future, with continued identification of novel bacteriophages, bacteriocins, prebiotics and probiotics, and anti-*Campylobacter* vaccine antigens. While the potentiality of these strategies to combat *Campylobacter* has been extensively investigated, their commercialization remains murky. In fact, several questions should be asked to determine the suitability of these approach for commercialization: For pre-harvest strategies: (a) can they provide consistent heterotypic protection? preferably reducing intestinal colony counts by at least 3 log_10_, (b) are they cost effective? (c) are they suitable for mass administration? (d) are they safe for chickens?. For post-harvest strategies: (a) can they consistently reduce *Campylobacter* load on poultry carcasses and eggshell? (b) do they have residual effects for humans?

More research is, indeed, needed to improve existing strategies or perhaps identify a novel strategy that meets the aforementioned criteria and more importantly, it should be industrially scalable and suitable for different commercial poultry systems in different countries. Future research should be directed at (a) identifying a novel and highly conserved immunogenic proteins that can induce cross-protective immunity against different strains of *C. jejuni*. Perhaps the innovative use of new technologies, such as reverse vaccinology, for prediction of novel antigenic targets could lead to development of multi-epitope vaccine capable of inducing cross-protection against different *Campylobacter* strains; (b) developing delivery systems for targeted delivery of vaccine formulations to the sites of *Campylobacter* colonization in chickens. The use of nanoparticles-based technologies for development of *Campylobacter* vaccines was found to be a promising replacement for older vaccine delivery methods. The uniquely tunable properties of nanoparticles enable them to be fine-tuned to be released in accordance with pathologic stimuli (pH, temperature, etc.) and thus can be specifically designed and engineered for targeted delivery of antigens to the immune inductive sites of the intestine (the sites of *Campylobacter* colonization; (c) developing a targeted competitive exclusion microbial consortium that possesses antagonistic ability against *Campylobacter*. Computational models can be employed to identify stable microbial communities with diverse immunomodulatory and anti-*Campylobacter* capabilities to be given to newly hatched chicks to establish a healthy and stable gut microbiome and prevent *Campylobacter* infection; and (d) developing a multifaceted approach that combines the previously mentioned strategies. 

Numerous studies have examined the combined effects of prebiotics and probiotics, however, to our knowledge, no study has been undertaken to investigate the combined effects of different feed additives and *Campylobacter* vaccines and feed additives. Implementing a combination strategy will likely result in synergistic effects, which may successfully reduce *Campylobacter* colonization of broilers to a degree where significant reductions in contamination of poultry products can be achieved. In addition to these measures, there is a need to increase public health awareness about proper handling of raw poultry meat and kitchen hygiene to minimize the risk of infection.

## Figures and Tables

**Figure 1 microorganisms-11-00113-f001:**
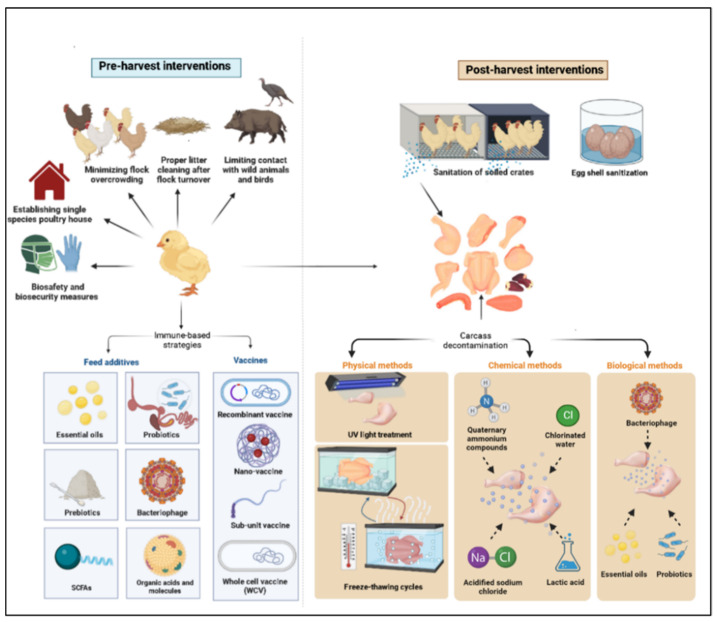
Pre- and post-harvest control measures.

**Figure 2 microorganisms-11-00113-f002:**
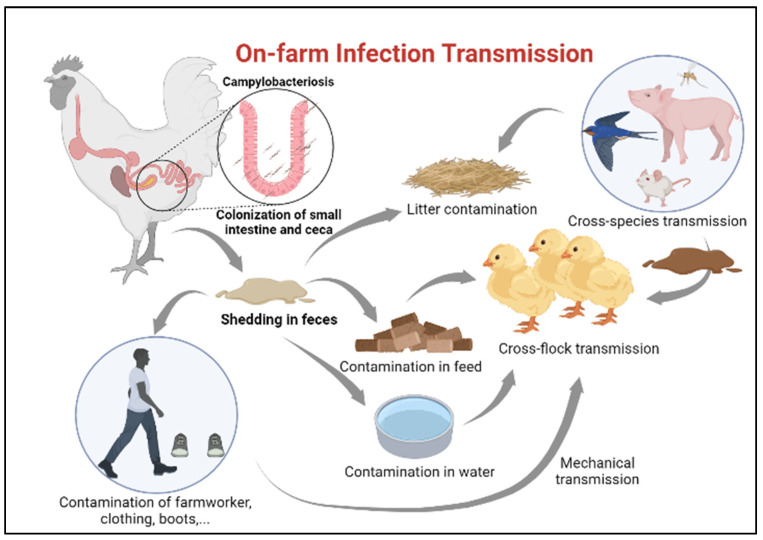
On-farm infection transmission.

**Table 1 microorganisms-11-00113-t001:** Vaccination trials for *C. jejuni* infection in poultry (modified from [49]).

Vaccine Type	Vaccine Active Components	Route of Administration	*C. jejuni* Challenge Strain/Dose per Bird	Effect on *Campylobacter* Colonization	Reference
Whole-cell vaccines	Whole cell (WC) and flagellin (*Fla*) or purified *Fla*	Intra-peritoneally (IP) followed by an IP or oral booster doses	*C. jejuni*, isolate #V2 (by mixing with seeder chickens)	WC+Fla (twice IP): Up to 2 log_10_ CFU/g WC+Fla (IP and orally) or Fla alone: No reduction	[45]
	Formalin inactivated WC with or without LT adjuvant	Esophageal gavage	*C. jejuni*, F1BCB (by mixing with seeder chickens)	WC alone: Up to 0.4 log_10_ on day 7 WC+LT: Up to 1.9 log_10_ on day 46	[46]
	Wild-type parental strain or the mutated strains (*CadF9, CiaB5, PldA23*, and DnaJA)	Intraperitoneal	*C. jejuni* F38011 (10^4^ CFU)	Up to 0.9 log_10_ CFU/g in *CiaB5* group only	[50]
	*C. jejuni* NCTC 11168 isogenic knockout mutants of *AhpC*, *KatA*, and *SodB*	Oral	*C. jejuni* NCTC 11168 (1 × 10^8^ CFU)	*ΔahpC* mutant: Up to 3 log_10_ *ΔkatA*: Up to 2 log_10_ *ΔsodB*: No effect on day 42 of age	[51]
Subunit vaccines	*C. jejuni* N-glycan with GlycoTag, or fused to the *E. coli* lipopolysaccharide-core	Oral	*C. jejuni*, 81–176 (10^2^ or 10^6^ CFU)	10 log_10_ reduction on day 35	[52]
	*FlaA/CadF/FlpA/CmeC* protein or *CadF-FlaA-FlpA* fusion protein	Intramuscular	*C. jejuni* F38011 (2 × 10^8^ CFU)	≥3 log_10_ on day 35	[53]
	Capsular polysaccharide conjugated to diphtheria toxoid CRM	Subcutaneous	*C. jejuni 81–176* (2 × 10^7^ CFU)	0.64 log_10_ CFU/g on day 38 of age	[54]
	*C. jejuni* outer membrane proteins	Subcutaneous/Oral	*C. jejuni*, 81–176 (1 × 10^8^ CFU)	SC: Below the detection limit (<10 CFU) Oral: No protection on day 42 of age	[55]
	*C. jejuni* Enterobactin-KLH conjugate	Intramuscular	*C. jejuni*, NCTC 11168 (1 × 10^4^ CFU)	>4 log10 on day 58 of age	[56]
Recombinant vaccines	*E. coli* expressing N-glycan protein with probiotics (A. mobilis DSM 15930 or *L. reuteri* CSF8)	Oral	*C. jejuni*, 81–176 (10^6^ CFU)	Up to 6 log_10_ on day 35	[57]
	*S. Typhimurium ΔaroA* mutant expressing *CjaA* as a plasmid-encoded fusion to fragment C of tetanus toxin	Subcutaneous /Oral	*C. jejuni*, MI (1 × 10^7^ CFU)	Oral: 1.4 log_10_ CFU/g Subcutaneous: 3.78 log_10_	[58]
	*Salmonella* strain carrying *C. jejuni* 72Dz/92 *CjaA* gene	Oral	*C. jejuni*, pUOA18 (2 × 10^8^ CFU)	6 log_10_ at 12 days post challenge	[59]
Nanoparticle-based vaccines	Chitosan/pCAGGS-flaA nanoparticles	Intranasal	*C. jejuni* ALM-80 (5 × 10^7^ CFU)	2 log_10_ CFU/g on day 35 of age	[60]
	PLGA-encapsulated *C. jejuni* outer membrane proteins	Subcutaneous /Oral	*C. jejuni* 81–176 (1 × 10^8^ CFU)	SC: Below the limit of detection Oral: No protection on day 42 of age	[55]
	Liposome encapsulated proteins (*CjaALysM* and *CjaDLysM*)	*In ovo*	*C. jejuni* 12/2 (10^6^ CFU)	2 log_10_ CFU/g on day 28 of age	[61]
	PLGA-encapsulated CpG ODN and *C. jejuni* lysate	Oral	*C. jejuni*, (81–176/ 10^7^ CFU)	Up to 2.4 log_10_ on day 37 of age	[62]

**Table 2 microorganisms-11-00113-t002:** Probiotics used to treat *Campylobacter* infections in chickens (modified from [100]).

Probiotic Strain	Type of Probiotics	Administration	*Campylobacter* Strain/Dose	Effect on *Campylobacter* Colonization	Reference
**Single strain**	*Escherichia coli* Nissle 1917	Two-weeks pre-harvest	Cocktail of six *C. jejuni* strains	Up to 2.6 log_10_ reduction	[101]
	*Lactobacillus salivarius* SMXD51	Administered at day 1 then every 2–3 days until 35 days orally	*C. jejuni* C97ANSES640 (1 × 10^4^ CFU)	0.8 log_10_ at 14 days and 2.81 log_10_ at 35 days	[102]
	*Lactobacillus plantarum* PA18A	Day 1 and 4 orally	*C. jejuni* strain 12/2 (1 × 10^4^ CFU)	1 log_10_ reduction	[103]
	*Lactobacillus gasseri* SBT2055 LG2055 *WT^CM^, Δapf1* and *Δapf2* mutant strains	Day 2–14 orally	*C. jejuni* 81–176 (1 × 10^6^ CFU)	*WT^CM^* and *Δapf2*: Up to 270-fold reduction *Δapf1*: No reduction	[104]
	*Lactobacillus gasseri* SBT2055	Day 2–14 orally	*C. jejuni* 81–176 (1 × 10^6^ CFU)	250-fold reduction	[105]
	*Lactobacillus**acidophilus NCFM* or *Lactobacillus crispatus* JCM5810 or *Lactobacillus gallinarum* ATCC or *Lactobacillus helveticus* CNRZ32	Day 1 and 4 orally	*C. jejuni* F38011 (1 × 10^8^ CFU)	Around 2 log_10_ reduction	[106]
	Calsporin^®^ (*Bacillus subtilis* C-3102) Ecobiol^®^ (*Bacillus amyloliquefaciens* CECT 5940)	Day 1 and 42 in feed	*C. jejuni* C97ANSES640 (1 × 10^4^ CFU)	Calsporin^®^: 0.25 log_10_ reduction on day 14 and 1.7 log_10_ on day 42 Ecobiol^®^: 1.12 log_10_ on day 35 and 1.2 log_10_ on day 42	[107]
	*Bacillus subtilis* DSM 17299 *or Saccharomyces cerevisiae boulardii*	Day 21–42 in feed	*C. jejuni* ST45 (1 × 10^4^ CFU)	*B. subtilis*: No reduction *S. cerevisiae*: Up to 0.3 log_10_ reduction	[108]
	*Bacillus* spp. (10 isolates individually tested)	Day 1 orally or intracloacally	*C. jejuni* cocktail of 4 strains (2.5 × 10^6^ CFU)	Intracloacally: 1–3 log_10_ Orally: 1 log_10_ for only 1 isolate	[109]
	Calsporin^®^ *(Bacillus subtilis* C-3102)	Day 1–42 in feed	Fecal contamination during processing	0.2 log_10_ reduction on chicken carcasses	[110]
	*Enterococcus faecalis* MB 5259	Day 1–21 orally	*C. jejuni* MB 4185 (KC 40) (2 × 10^4^ CFU)	0.4 log_10_ in only one of the groups received 10^4^ CFU *E. faecalis* No reduction in the chickens received 10^8^ CFU *E. faecalis*	[111]
	*Enterococcus faecium* NCIMB 11508	Day 1 and 28 orally	Naturally infected	No reduction in the relative abundance of *Campylobacter*	[112]
	Microencapsulated *Bifidobacterium longum* PCB133 + oligosaccharides	Day 1–14 in feed	Naturally infected	Up to 1.4 log_10_	[113]
	*Bifidobacterium longum* PCB 133	Day 1–15 intraesophageally	Naturally infected	1 log_10_ reduction	[114]
**Multi-strain**	Avian Pac Soluble (*Lactobacillus acidophilus + Streptococcus faecium*)	Day 1–3 in drinking water	*C. jejuni* C101 (2.7 × 10^4^ CFU)	Two-thirds reduction in *C. jejuni* shedding	[115]
	*Bacillus* spp.+ *Lactobacillus salivarius subsp. salivarius + L. salivarius sub sp. salicinius*	Day 1 orally	*C. jejuni* cocktail of 4 strains (2.5 × 10^6^ CFU)	1–2 log_10_ in only one of 3 trials	[116]
	PrimaLac *(Lactobacillus acidophilus + Lactobacillus casei + Bifidobacterium thermophilus + Enterococcus faecium)*	Day 1–42 in feed	Naturally infected	12% reduction of *C. jejuni* presence	[117]
	K-bacteria + competitive exclusion Broilact ^®^	Day 1–38 in drinking water	*C. jejuni* T23/42 (1.3 × 10^4^ CFU)	Up to 2 log_10_	[118]
	*PoultryStar sol* ^®^ *(Enterococcus faecium + Pediococcus acidilactici + Bifidobacterium animalis + Lactobacillus salivarius + Lactobacillus reuteri)*	Day 1–15 in drinking water	*C. jejuni* 3015/2010 (1 × 10^4^ CFU)	≥ 6 log_10_	[119]
	*Lactobacillus paracasei J.R + Lactobacillus rhamnosus 15b + Lactococcus lactis Y + Lactococcus lactis FOa*	Day 1–42 in drinking water	Naturally infected	Up to 5 log_10_	[120]
	Lavipan (multispecies probiotic): *Lactococcus lactis* IBB 500, *Carnobacterium divergens* S-1, *Lactobacillus casei* OCK 0915, L0915, *L. plantarum* OCK 0862, and *Saccharomyces cerevisiae* OCK 0141	Day 1–37 in feed	Naturally infected	<1 log_10_	[121]
	*Citrobacter diversus* 22 + *Klebsiella pneumonia* 23 + *Escherichia coli* 25 + mannose	Day 1 and 3 orally	*C. jejuni* 10^8^ (1 × 10^8^ CFU)	Up to 70 % reduction	[122]

**Table 3 microorganisms-11-00113-t003:** Bacteriocins used for the treatment of *Campylobacter* colonization in poultry (modified from [49]).

Bacteriocins Source and Name	Dose and Duration of Administration	*C. jejuni* Strain and Dose	Effect on *Campylobacter* Colonization	References
*Enterococcus faecium* (E 50–52)	31.2 mg/kg of feed Day 4–7 of age	10^6^ CFU *C. jejuni* isolates B1 and L4 on day of hatch	<10^2^ CFU/g reduction on day 15 of age	[129]
	12.5 mg/liter of drinking water Day 35–41 of age	Environmentally infected	Below the limit of detection on days 40 and 41 of age	
*Enterococcus durans*/*faecium*/*hirae* (E-760)	31.2 mg/kg of feed Day 4–7 of age	10^6^ CFU *C. jejuni* isolates B1 and L4 on day of hatch	Below the limit of detection on day 7 of age	[130]
	125 mg/kg of feed	Naturally colonized	Below the limit of detection on day day 43 of age	
*Lactobacillus* *Salivarius* (OR-7)	250 mg/kg of feed Day 7–9 of age	10^8^ CFU *C. jejuni* strain AL-22 or BH-6 or CL-11 on day 1 of age	Below the limit of detection on day 10 of age	[126]
*Paenibacillus polymyxa* (B602), or *Lactobacillus salivarius* (OR7)	250 mg/kg of feed on day 10–12 of age (turkey poults)	10^6^ CFU of a mixture of 3 *C. coli* isolates on day 3 of age	Below the limit of detection on day on day 12 of age	[127]
Microencapsulated *Paenibacillus polymyxa* (B-30509), or *Lactobacillus. salivarius* (B-30514)	250 mg/kg of feed	6 × 10^6^ CFU on day 1–4 of age	*P. polymyxa* B-30509: complete elimination *L. salivarius* B-30514: <1 log_10_ on day 7 of age	[131]
*Paenibacillus polymyxa* microencapsulated (SRCAM 602)	250 mg/kg of feed on day 7–9 of age	10^8^ CFU *C. jejuni* strain AL-22 or BH-6 or CL-11 on day 1 of age	Below the limit of detection on day 10 of age	[128]
*Enterococcus faecium* (E-760) or (E-760 E- resistant mutants (JL341, K58, or JL106))	5 mg/kg body weight/day orally on day 9 for 3 consecutive days	10^7^ CFU *C. jejuni* NCTC 11168 on day 2 of age	Slightly reduced on days 24 and 44 of age	[132]

## Data Availability

Not applicable.

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
