# Peer review of "Intervention Strategies to Control Campylobacter at Different Stages of the Food Chain"

_microorganisms, 2023, doi:10.3390/microorganisms11010113_

Round 1

Reviewer 1 Report

The manuscript aims to provide a review of interventions to control Campylobacter at different stages of the food chain. I think the remit of the review is too broad and general and it would benefit with more focus on specific outputs. The studies reviewed have many different outputs and is commended for that however the review lacks a clear narrative with interpretation of which interventions are truly noteworthy with regard to control of Campylobacter in the food chain.

If the aim of the review is to look broadly at options for controlling Campylobacter and reducing risk then interventions such education of operatives on biosecurity should be considered, enhanced biosecurity options, partial depopulation practices, schedule and logistic slaughter, processing refinements, packaging innovations and end consumer education. 

When studies on control of farm colonisation are presented, it should be with a focus on the impact of the number of birds that are entering abattoir carrying Campylobacter, and what level of campylobacter is being carried. These are going to be the major risks leading to contamination of the product during processing.

Looking at studies of post-harvest stages, I think the focus should be most on the actual reductions in the numbers of contaminated carcases and the level of contamination and less focussed on the estimates of risk-reduction to people. I think it is challenging to look at potentially different outputs alongside each other unless more interpretation is given. There are some studies presented here that described treatments that can reduce a level of contamination by 5 log10, from 8log10 to 3log10. That may be a significant log reduction, but is it realistic? A carcase with 3log10/g of skin is still considered a high-risk carcase in some contexts? 

Many of the intervention strategies reviewed here, sound on first reading to be promising, but then the reader will ask, if it this good why is not being used? Is it cost, user acceptance, toxic, etc., this is missing in many parts of the review.

In the conclusions and future prospects, the options for future research are very narrow banking on vaccine and competitive exclusion are these really the only areas that merit any further research?

More comments on the manuscript are in the attached.

Author Response

We thank the reviewer for the thorough review. Your comments and suggestions were very constructive and useful in shaping the current version of our manuscript. We have addressed all your concerns raised here and in the attachment. We used the "track change" function so that the changes are made clear for the second round of review.

Reviewer: I think the remit of the review is too broad and general and it would benefit with more focus on specific outputs. The studies reviewed have many different outputs and is commended for that however the review lacks a clear narrative with interpretation of which interventions are truly noteworthy with regard to control of Campylobacter in the food chain

Authors: We thank the reviewer for the thorough review. In the new version, we provided a clear interpretation on the success and failure the investigated approaches and challenges of their commercialization. We have also added new sections to the review and made extensive changes to the previous version following the reviewer’s suggestion.

Reviewer: When studies on control of farm colonisation are presented, it should be with a focus on the impact of the number of birds that are entering abattoir carrying Campylobacter, and what level of campylobacter is being carried. These are going to be the major risks leading to contamination of the product during processing.

Looking at studies of post-harvest stages, I think the focus should be most on the actual reductions in the numbers of contaminated carcases and the level of contamination and less focussed on the estimates of risk-reduction to people. I think it is challenging to look at potentially different outputs alongside each other unless more interpretation is given. There are some studies presented here that described treatments that can reduce a level of contamination by 5 log10, from 8log10 to 3log10. That may be a significant log reduction, but is it realistic? A carcase with 3log10/g of skin is still considered a high-risk carcase in some contexts? 

Authors: As per your suggestions, we have added the following sections to the current version of the review:

On-farm control measures alone have not been sufficient to eliminate Campylobacter in poultry. Sanitation practices in poultry processing facilities should also be implemented to further reduce Campylobacter levels at later stages of the food supply chain. In a recent report by European Food Safety Authority (EFSA), the proportion of broiler flocks infected by Campylobacter varies widely (ranging from 2 to 100%), and strongly correlates with the prevalence of Campylobacter on broiler carcasses (4.9% to 100%). Chickens carry a high load of Campylobacter of approximately 8 log10 CFU/g in their caeca prior to slaughter. Contamination of chickens’ feathers with fecal material during transportation to the slaughterhouse can also be a significant external source of carcass contamination during the plucking/defeathering process.

Even with measures taken for cleaning and disinfecting transportation crates, Campylobacter was detected in 57% of the swab samples collected from cleaned crates and a notable increase in the number of infected chickens by 9% was also observed in the cloacal swabs following transportation, but whether the increase in the number of infected birds enhanced the risk of carcass contamination during processing was not investigated in this study.

As such, increasing down time of crates between flocks and effective cleaning could put in place to lower the risk of horizontal exposure. In addition to these measures, effective carcass decontamination practices should be considered to reduce Campylobacter concentration in the poultry meat. Contamination of meat products by gut contents is difficult to prevent during processing at slaughter plants because of the high numbers of C. jejuni in the gut, and the large percentage of birds infected. Data from surveillance studies in different countries indicated a high prevalence of Campylobacter on raw retail chicken meat. For example, Campylobacter was detected in 28.6%, 36.5%, 41.2% and 52.2%, 59.9% of samples from chicken meat from retail stores in the United Arab Emirates, Qatar, the United Kingdom, Saudi Arabia, and Canada, respectively. It is estimated that a reduction of Campylobacter counts in the neck and breast skin to 103 CFU/g reduces the public health risk by 50%. The following section summarizes the physical, chemical, and biological control measures that have been, or are being taken, to reduce Campylobacter load on poultry carcasses.

We also added a section about Sanitation in slaughter plants

In addition to the risk of carcass contamination during evisceration, cross contamination from contaminated processing equipment surfaces, due to insufficient cleaning and disinfection, should also be considered as another source of carcass contamination during the slaughter process. Sounmet and Sanders were the first to report the survival of Campylobacter on cleaned and disinfected surfaces of four French slaughterhouses. It is, however, unclear, whether the transportation crates or the previously slaughter flock was responsible for the observed contamination, since the same Campylobacter strain was isolated from the crates and chicken carcasses. These data highlight the importance of carcass treatment at the end of the process line.

Reviewer: Many of the intervention strategies reviewed here, sound on first reading to be promising, but then the reader will ask, if it this good why is not being used? Is it cost, user acceptance, toxic, etc., this is missing in many parts of the review.

Authors: This is a very good question. In this review, we provided the up-to-date information on various intervention strategies, and we also provided some explanation on why the effective strategies are not commercialized. We attributed this to many reasons which were all discussed in the manuscript and provided in the inclusion. For example, a) regulatory hurdles or b) For organic acids: studies lack data about their effect on growth performance for example, we add the following segment under the organic acid “Nonetheless, this study did not assess the effects of these organic acids on feed intake and growth performance of chickens. Although their undeniable beneficial effects, it is important to note that inclusion of these organic acids in poultry feed may affect its palatability, thereby reducing chicken feed intake. Regardless of these potential limitations, the effectiveness of organic acids appears to be largely dependent on the type, concentration, and combination of the organic acid used c) large scale trials are needed to validate the obtained results or d) most of these strategies has not been texted in heterologous challenge models, extra.

We have also added the following section to the conclusion: While the potentiality of these strategies to combat Campylobacter has been extensively investigated, their commercialization remains murky. In fact, several questions should be asked to determine the suitability of these approach for commercialization: a) can they provide heterotypic protection? preferably reducing intestinal colony counts by at least 3 log10? b) are they cost effective? c) are they suitable for mass administration? d) are they safe for chickens and humans? and e) do they have residual effects for humans?

Reviewer: In the conclusions and future prospects, the options for future research are very narrow banking on vaccine and competitive exclusion are these really the only areas that merit any further research?

Authors: We have expanded the conclusion. We also welcome any input from the reviewer.

More research is, indeed, needed to improve existing strategies or perhaps identify a novel strategy that meets the aforementioned criteria and more importantly, it should be industrially scalable and suitable for different commercial poultry systems in different countries. Future research should be directed at a) identifying a novel and highly conserved immunogenic proteins that can induce cross-protective immunity against different strains of C. jejuni. Perhaps the innovative use of new technologies, such as reverse vaccinology, for prediction of novel antigenic targets could lead to development of multi-epitope vaccine capable of inducing cross-protection against different Campylobacter strains. b) developing delivery systems for targeted delivery of vaccine formulations to the sites of Campylobacter colonization in chickens. The use of nanoparticles-based technologies for development of Campylobacter vaccines was found to be a promising replacement for older vaccine delivery methods. The uniquely tunable properties of nanoparticles enable them to be fine-tuned to be release in accordance with pathologic stimuli (pH, temperature, etc.) and thus can be specifically designed and engineered for targeted delivery of antigens to the immune inductive sites of the intestine (the sites of Campylobacter colonization c) developing a targeted competitive exclusion microbial consortium that possesses antagonistic ability against Campylobacter. Computational models can be employed to identify stable microbial communities with diverse immunomodulatory and anti-Campylobacter capabilities to be given to newly hatched chicks to establish a healthy and stable gut microbiome and prevent Campylobacter infection d) developing a multifaceted approach that combines the previously mentioned strategies. Numerous studies have examined the combined effects of prebiotics and probiotics, however, to our knowledge, no study has been undertaken to investigate the combined effects of Campylobacter vaccines and feed additives. Implementing a combination strategy will likely result in synergistic effects, which may successfully reduce Campylobacter colonization of broilers to a degree where significant reductions in contamination of poultry products can be achieved. In addition to these measures, there is a need to increase public health awareness about proper handling of raw poultry meat and kitchen hygiene to minimize the risk of infection.

Reviewer 2 Report

  • I think that the review, as currently configured, needs substantial reframing prior to publication. It is clear that there are certain areas that the authors are much more familiar with. The authors have done an excellent job in covering the more experimental/speculative areas for Campylobacter control (vaccines, probiotics, prebiotics, phage, etc). The major issue I have is that in framing their review as all-encompassing (i.e. farm to fork) it becomes apparent that they are not nearly as familiar with on-farm measures. As such, they make no distinction between modern state-of-the-art production systems and more resource-constrained production systems that may be unable to implement some of the measures that they discuss. So when the authors advocate for improved on-farm biosecurity measures and improved production practices, this ignores what is already being done in many countries. And yet, Campylobacter remains a challenge despite the fact that such measures are being undertaken, so it is clear that although these measures have an effect, it is not enough. In my opinion, this is the context that they should utilize in framing their review of experimental control measures (i.e. these novel measures may be needed because existing measures are not sufficient to control the problem). The authors seem to suggest that the problem lies with the insufficient implementation of on-farm mitigation measures. I would suggest that they examine production systems in countries like the UK and New Zealand, which have essentially implemented these measures fully and which still face substantial Campylobacter problems. My suggestion would be to reframe the review as focusing on the experimental areas for control given the fact that state-of-the-art measures being undertaken with regard to biosecurity and other on-farm mitigation strategies appear to be insufficient. Personally, I will remain skeptical of the experimental measures until they are tested on the "real" breadth of genetic diversity circulating in the wild rather than on one or small cocktails of laboratory strains. At the same time, I don't have to agree with the authors' thesis to deem the review as essential reading when it comes to covering the experimental control measures. And I think that they have done an excellent on that front.
Minor
  • L48: there are much more recent estimates for the economic costs in the USA associated with campylobacteriosis by the Economic Research Service of the USDA
  • L56: colonization of chicks upon entry into a broilerhouse is a very critical part of the process. Accordingly, reducing this to a single citation seems like something that you should reconsider in favor of a more comprehensive review of literature focusing strictly on this aspect of the story.
  • L65: to my knowledge, there is scant evidence for transmission of Campylobacter via eggs/eggshells. I found its inclusion here extremely surprising.
  • L71: should be plural "carcasses"
  • L92-94:  The citation you include is over 30 years old and much has changed in terms of production systems. So that entire sentence is incorrect and not a reflection of the current state of knowledge. At the very least, there is deep decontamination between production cycles and many farms now allow a barn to remain unused between production cycles as a further added measure. In truth, given the decontamination and strict biosecurity measures in most farms, it remains a bit of a mystery how Campylobacter enters broiler barns in the first place. Significant transmission between flocks may have occurred 30 years ago, not now. This sentence trivializes the many efforts that poultry producers are taking to reduce the incidence of contaminated flocks.
  • L133-135: this is the commonly accepted timeline to flock colonization. And yet this contrasts with what you said in L56-57. Chickens are typically harvested within 5 weeks of hatching. So colonization 2-3 weeks post-hatching would not be considered "shortly after hatching".

Author Response

Reviewer: As such, they make no distinction between modern state-of-the-art production systems and more resource-constrained production systems that may be unable to implement some of the measures that they discuss. So when the authors advocate for improved on-farm biosecurity measures and improved production practices, this ignores what is already being done in many countries. And yet, Campylobacter remains a challenge despite the fact that such measures are being undertaken, so it is clear that although these measures have an effect, it is not enough. In my opinion, this is the context that they should utilize in framing their review of experimental control measures (i.e. these novel measures may be needed because existing measures are not sufficient to control the problem). The authors seem to suggest that the problem lies with the insufficient implementation of on-farm mitigation measures. I would suggest that they examine production systems in countries like the UK and New Zealand, which have essentially implemented these measures fully and which still face substantial Campylobacter problems. My suggestion would be to reframe the review as focusing on the experimental areas for control given the fact that state-of-the-art measures being undertaken with regard to biosecurity and other on-farm mitigation strategies appear to be insufficient. Personally, I will remain skeptical of the experimental measures until they are tested on the "real" breadth of genetic diversity circulating in the wild rather than on one or small cocktails of laboratory strains. At the same time, I don't have to agree with the authors' thesis to deem the review as essential reading when it comes to covering the experimental control measures. And I think that they have done an excellent on that front.

Authors: We thank the reviewer for the valuable input. We have reframed the biosecurity measures as per your suggestions and added the following sections that cover the different production systems.

Campylobacter transmission can also be attributed to human activity on poultry farms; contaminated clothes, skin, and boots of farmworkers and transport crates, either due to biosecurity breaches during the process of partial flock depopulation (thinning) or insufficient implementation of strict biosecurity measures, may contribute to the transmission of Campylobacter from the external environment into poultry houses. Partial flock depopulation is a common practice in many European countries where a portion of a flock is removed and sent for slaughtering before the final slaughter age. This process was found to be associated with an increased risk of Campylobacter introduction into the broiler house. In a recent survey in Ireland, an increase in prevalence of Campylobacter infection was observed due to the thinning process, where Campylobacter was detected in 38% of the neck skin samples of chickens removed early for slaughtering compared to 67% in the remainder flock.

It is widely believed that raising chicken in modern state-of-the-art production systems and full implementation of biosecurity measures would significantly tackle Campylobacter infections. Yet, Campylobacter remains a challenge despite the fact that such measures are being undertaken in European countries, like the UK and New Zealand. As a result, novel on-farm mitigation strategies and complementary approaches should be applied.

We also added in the post-harvest section that on-farm control measures alone have not been sufficient to eliminate Campylobacter in poultry. Sanitation practices in poultry processing facilities should also be implemented to further reduce Campylobacter levels at later stages of the food supply chain. In a recent report by European Food Safety Authority (EFSA), the proportion of broiler flocks infected by Campylobacter varies widely (ranging from 2 to 100%), and strongly correlates with the prevalence of Campylobacter on broiler carcasses (4.9% to 100%). Chickens carry a high load of Campylobacter of approximately 8 log10 CFU/g in their caeca prior to slaughter. Contamination of chickens’ feathers with fecal material during transportation to the slaughterhouse can also be a significant external source of carcass contamination during the plucking/defeathering process.

Minor

  • L48: there are much more recent estimates for the economic costs in the USA associated with campylobacteriosis by the Economic Research Service of the USDA

We added the following statement: According to the latest USDA-economic research service report, the annual healthcare costs incurred by foodborne diseases in the US are estimated to be about $15 billion, of which $1.6 billion caused by Campylobacter species

  • L56: colonization of chicks upon entry into a broilerhouse is a very critical part of the process. Accordingly, reducing this to a single citation seems like something that you should reconsider in favor of a more comprehensive review of literature focusing strictly on this aspect of the story.

We have added more citations

  • L65: to my knowledge, there is scant evidence for transmission of Campylobacter via eggs/eggshells. I found its inclusion here extremely surprising.

We have revised this part and here what we added to the current version.

While the horizontal route plays a major role in Campylobacter transmission, vertical transmission is unlikely. A number of studies have reported infrequent detection of C. jejuni in chicken eggs; however, whether it is due to internal eggshell contamination from the hen’s reproductive tract or external eggshell contamination with feces of infected chicken remains controversial. In an experimental trial, C. jejuni was detected in 20% of Specific Pathogen Free eggs (SPF) 3 hours following the exposure to C. jejuni-contaminated wood shavings. In another study, inoculation of  C. jejuni into the air space of embryonated SPF eggs resulted in high embryonic mortality, with 87% of embryos died on the second day post incubation [38]. Nonetheless, there is as yet no evidence whether heavy contamination of eggshells with Campylobacter from environmental sources would lead to similar outcomes. 

  • L71: should be plural "carcasses"

Thank you for the thorough review. We corrected it.

  • L92-94:  The citation you include is over 30 years old and much has changed in terms of production systems. So that entire sentence is incorrect and not a reflection of the current state of knowledge. At the very least, there is deep decontamination between production cycles and many farms now allow a barn to remain unused between production cycles as a further added measure. In truth, given the decontamination and strict biosecurity measures in most farms, it remains a bit of a mystery how Campylobacter enters broiler barns in the first place. Significant transmission between flocks may have occurred 30 years ago, not now. This sentence trivializes the many efforts that poultry producers are taking to reduce the incidence of contaminated flocks.

We have removed many of the old citations and addressed your concerns in the current version. We added the following section and some other new sections to this version (please see the track changes in the attachment):

Increased turnaround periods have also been reported to reduce the risk of new flocks becoming colonized. This is largely because Campylobacter is increasingly less effective at colonization with greater periods spent outside the host gastrointestinal tract. Lazaro et al. have reported that Campylobacter can survive up to 7 months in a viable but not culturable state. As such, increasing down time of crates between flocks and effective cleaning could put in place to lower the risk of horizontal exposure. In addition to these measures, effective carcass decontamination practices should be considered to reduce Campylobacter concentration in the poultry meat. Contamination of meat products by gut contents is difficult to prevent during processing at slaughter plants because of the high numbers of C. jejuni in the gut, and the large percentage of birds infected. Data from surveillance studies in different countries indicated a high prevalence of Campylobacter on raw retail chicken meat. For example, Campylobacter was detected in 28.6%, 36.5%, 41.2% and 52.2%, 59.9% of samples from chicken meat from retail stores in the United Arab Emirates, Qatar, the United Kingdom, Saudi Arabia, and Canada, respectively. It is estimated that a reduction of Campylobacter counts in the neck and breast skin to 103 CFU/g reduces the public health risk by 50%. The following section summarizes the physical, chemical, and biological control measures that have been, or are being taken, to reduce Campylobacter load on poultry carcasses.

  • L133-135: this is the commonly accepted timeline to flock colonization. And yet this contrasts with what you said in L56-57. Chickens are typically harvested within 5 weeks of hatching. So colonization 2-3 weeks post-hatching would not be considered "shortly after hatching".

We agree with the reviewer and have made the necessary correction throughout the manuscript.

Reviewer 3 Report

This is a comprehensive review of the literature around different intervention strategies to control campylobacter.

However I was disappointed that after 700 lines of literature review, the conclusions and future prospects was a paltry 19 lines.  the future research suggestions lack any justification of why those are the best to proceed with. I Missing is consideration of the cost and practicality of implementing any of these control measures. Cost can be difficult, but a relative cost would be useful. How easy are these to scale up to millions of chickens?  More discussion to compare and contrast approaches. A number of log removals are identified. Some consideration of the impact different log levels of reduction would have would be nice to see.  Do we need 6 log removal or would 2 log make a significant difference?  Consideration of application in different countries?

The authors have done the hard work finding and reviewing an extensive bit of literature. But think need to pull it all together to actually meet the statement in abstract of providing insights in optimization of these approaches. 

Table 1. There are studies in Table 1 that aren't referred to in the text (eg 43), and studies in text that aren't in Table 1 (eg 51, 52)/Think can remove the heading Exposure to infected birds.  5th column needs consistent wording to enable comparisons.  Some say reduction, others just "up to 2 logs".  What does significant reduction mean?  Can you convert to logs?

Table 2 also needs sorting or ordering into logical order. Group similar probiotics.  Aso think need to clarify genus when have more than one with same letter.

Line 774.  I think this is wrong. Don't think there is evidence in those references for 40-60% of campy to be caused by campy on shells of eggs!

Minor comments

Line 19 and 46:  most REPORTED, no reportable.

Line 40. Delete "in severe cases"

Line 88-89. Delete "There are different", "various", and "these measures"

Author Response

Reviewer: However I was disappointed that after 700 lines of literature review, the conclusions and future prospects was a paltry 19 lines.  the future research suggestions lack any justification of why those are the best to proceed with. I Missing is consideration of the cost and practicality of implementing any of these control measures. Cost can be difficult, but a relative cost would be useful. How easy are these to scale up to millions of chickens?  More discussion to compare and contrast approaches. A number of log removals are identified. Some consideration of the impact different log levels of reduction would have would be nice to see.  Do we need 6 log removal or would 2 log make a significant difference?  Consideration of application in different countries?

Authors: We thank the reviewer for the thorough review and valuable input. We have addressed all your concerns in the current version and below is the new conclusion. We also welcome any further input from the reviewer.

The continued increase in the incidence of human campylobacteriosis, which is estimated to be increased by 70% in 2018 from 2006 data, and associated healthcare costs necessitate an urgent need for effective ways to combat Campylobacter infection in poultry and prevent its transmission to humans through contaminated poultry products. Although there is no effective intervention measure available to “completely” eliminate Campylobacter in poultry, there is a considerable amount of promise for the future, with continued identification of novel bacteriophages, bacteriocins, prebiotics and probiotics, and anti-Campylobacter vaccine antigens. While the potentiality of these strategies to combat Campylobacter has been extensively investigated, their commercialization remains murky. In fact, several questions should be asked to determine the suitability of these approach for commercialization: a) can they provide heterotypic protection? preferably reducing intestinal colony counts by at least 3 log10? b) are they cost effective? c) are they suitable for mass administration? d) are they safe for chickens and humans? and e) do they have residual effects for humans?

More research is, indeed, needed to improve existing strategies or perhaps identify a novel strategy that meets the aforementioned criteria and more importantly, it should be industrially scalable and suitable for different commercial poultry systems in different countries. Future research should be directed at a) identifying a novel and highly conserved immunogenic proteins that can induce cross-protective immunity against different strains of C. jejuni. Perhaps the innovative use of new technologies, such as reverse vaccinology, for prediction of novel antigenic targets could lead to development of multi-epitope vaccine capable of inducing cross-protection against different Campylobacter strains. b) developing delivery systems for targeted delivery of vaccine formulations to the sites of Campylobacter colonization in chickens. The use of nanoparticles-based technologies for development of Campylobacter vaccines was found to be a promising replacement for older vaccine delivery methods. The uniquely tunable properties of nanoparticles enable them to be fine-tuned to be release in accordance with pathologic stimuli (pH, temperature, etc.) and thus can be specifically designed and engineered for targeted delivery of antigens to the immune inductive sites of the intestine (the sites of Campylobacter colonization c) developing a targeted competitive exclusion microbial consortium that possesses antagonistic ability against Campylobacter. Computational models can be employed to identify stable microbial communities with diverse immunomodulatory and anti-Campylobacter capabilities to be given to newly hatched chicks to establish a healthy and stable gut microbiome and prevent Campylobacter infection d) developing a multifaceted approach that combines the previously mentioned strategies. Numerous studies have examined the combined effects of prebiotics and probiotics, however, to our knowledge, no study has been undertaken to investigate the combined effects of Campylobacter vaccines and feed additives. Implementing a combination strategy will likely result in synergistic effects, which may successfully reduce Campylobacter colonization of broilers to a degree where significant reductions in contamination of poultry products can be achieved. In addition to these measures, there is a need to increase public health awareness about proper handling of raw poultry meat and kitchen hygiene to minimize the risk of infection.

Table 1. There are studies in Table 1 that aren't referred to in the text (eg 43), and studies in text that aren't in Table 1 (eg 51, 52)/Think can remove the heading Exposure to infected birds.  5th column needs consistent wording to enable comparisons.  Some say reduction, others just "up to 2 logs".  What does significant reduction mean?  Can you convert to logs?

We have made the necessary changes and added more studies to the table.

Table 2 also needs sorting or ordering into logical order. Group similar probiotics.  Aso think need to clarify genus when have more than one with same letter.

We have made the necessary changes. Thank you.

Line 774.  I think this is wrong. Don't think there is evidence in those references for 40-60% of campy to be caused by campy on shells of eggs!

We agree with the reviewer. We have decided to delete it.

Minor comments

Line 19 and 46:  most REPORTED, no reportable.

Corrected

Line 40. Delete "in severe cases"

Deleted

Line 88-89. Delete "There are different", "various", and "these measures"

Deleted

Round 2

Reviewer 3 Report

Changes made to my satisfaction.